# PIF7 is a master regulator of thermomorphogenesis in shade

Yogev Burko [1,2,3] ✉, Björn Christopher Willige [2,3], Adam Seluzicki[2,3], Ondřej Novák[4,5], Karin Ljung [5] & Joanne Chory [2,3] ✉

The size of plant organs is highly responsive to environmental conditions. The plant's embryonic stem, or hypocotyl, displays phenotypic plasticity, in response to light and temperature. The hypocotyl of shade avoiding species elongates to outcompete neighboring plants and secure access to sunlight. Similar elongation occurs in high temperature. However, it is poorly understood how environmental light and temperature cues interact to effect plant growth. We found that shade combined with warm temperature produces a synergistic hypocotyl growth response that dependent on PHYTOCHROME-INTERACTING FACTOR 7 (PIF7) and auxin. This unique but agriculturally relevant scenario was almost totally independent on PIF4 activity. We show that warm temperature is sufficient to promote PIF7 DNA binding but not transcriptional activation and we demonstrate that additional, unknown factor/s must be working downstream of the phyB-PIF-auxin module. Our findings will improve the predictions of how plants will respond to increased ambient temperatures when grown at high density.

Plants evolved with a unique flexibility in regulating their growth in response to environmental changes. Just after germination, they rapidly elongate the embryonic stem (hypocotyl) to emerge from the soil and capture light as fast as possible. Once they sense light, the hypocotyl elongation rate decreases, ensuring a stable structure to support the mature plant. However, they can increase the hypocotyl elongation rate again to outcompete surrounding vegetation[1–3]. In this process, termed the shade avoidance response, plants can detect other plants' proximity by perceiving a change in the ratio of Red to Far-Red (R/FR) light due to the reflected FR light from neighboring plants[4]. The hypocotyl also elongates in response to increased ambient temperatures, referred to as thermomorphogenesis[5].

In recent years, many studies combined different environmental conditions to mimic the natural environment and examine effects on growth. It was found that shade (low R/FR) and temperature strongly interact[6–10], and the combination of low R/FR and warm temperature

resulted in longer plants than either of the single conditions[7,10]. This interaction is particularly important given the predicted increase in ambient temperature due to global warming and that crops growing in field conditions are likely to sense both signals simultaneously.

Current models suggest that the responses to shade and to warm ambient temperatures are sensed and executed by common molecular factors. In *Arabidopsis thaliana*, both stimuli are perceived by the phytochrome B (phyB) photoreceptor. In the shade, the low R/FR ratio converts phyB to its inactive form. Warm ambient temperature increases the rate of phyB's spontaneous reversion from its active (Pfr) to its inactive (Pr) form. This occurs mainly during nighttime and increases the pool of inactive phyB over the total phyB pool[11–14]. Inactivating phyB relieves the repression of the PHYTOCHROME-INTERACTING FACTOR (PIF) transcription factors, allowing them to promote transcription of growth-promoting target genes. PIF4 and PIF7 have been shown to be the dominant proteins regulating

[1]Institute of Plant Sciences, ARO, Volcani Institute, HaMaccabbim Road 68, Rishon LeZion 7505101, Israel. [2]Howard Hughes Medical Institute, Salk Institute for Biological Studies, La Jolla, CA 92037, USA. [3]Plant Biology Laboratory, Salk Institute for Biological Studies, 10010 North Torrey Pines Road, La Jolla, CA 92037, USA. [4]Laboratory of Growth Regulators, Faculty of Science, Palacký University and Institute of Experimental Botany, The Czech Academy of Sciences, Šlechtitelů 27, CZ-78371 Olomouc, Czech Republic. [5]Umeå Plant Science Centre, Department of Forest Genetics and Plant Physiology, Swedish University of Agricultural Sciences, SE-901 83 Umeå, Sweden. ✉e-mail: yogevb@volcani.agri.gov.il; chory@salk.edu

thermomorphogenesis and shade avoidance response, respectively. PIF4 has been shown to accumulate at both the mRNA and protein levels during thermomorphogenesis. While the precise mechanisms regulating PIF4 transcript levels during thermomorphogenesis are not clear yet, its abundance is regulated by several proteins, including CONSTITUTIVELY PHOTOMORPHOGENIC 1 (COP1), DE-ETIOLATED 1 (DET1), SUPPRESSOR OF PHYA-105 1 (SPA1), and HEMERA (HMR)[15–18]. Recent studies also revealed a major role for PIF7 during thermomorphogenesis[19,20]. Warm ambient temperature increases PIF7 protein abundance by enhancing translation via alternative hairpin formation in the 5′ untranslated region of the PIF7 mRNA[20]. In response to shade, PIF7 protein activity is regulated by its phosphorylation state[21–23]. In addition, only the triple mutant pif457 completely blocks the elongation responses to low R/FR conditions[23,24], and PIF7 activity may be dependent on PIF4[19]. Therefore, the roles of redundancy and co-regulation among the PIFs during these responses are still unclear.

Auxin pathway genes are highly responsive to low R/FR or warm ambient temperature treatments[25–27]. Both signals are perceived in the cotyledons and young leaves and increase the expression of YUCCA (YUC) genes that encode auxin biosynthetic enzymes. Consequently, auxin accumulates and is transported into the hypocotyl, leading to the induction of cell elongation[26–29]. This process is dependent on PIF binding to and transcriptional activation of YUCs. In agreement with this, the yuc2589 mutant is insensitive to shade or warm ambient temperature[19,30]. Other hormones, including brassinosteroids, gibberellin, and ethylene have all been shown to be part of shade- or warm temperature-induced hypocotyl growth, and all interact with auxin signaling during these responses[5,25,27,29,31–34].

Here we compare the responses to low R/FR and warm ambient temperatures applied individually and simultaneously. This comparison allowed us to identify crucial differences between the shade and warm temperature responses, which were previously thought to be very similar. Furthermore, we identified PIF7 as the dominant player during the simultaneous response to low R/FR and warm temperature. Surprisingly, this unique but agriculturally relevant scenario was almost totally independent on PIF4 activity.

## Results

### Synergistic effect of low R/FR and elevated temperature on hypocotyl growth

Previous studies suggest that both low R/FR and warm temperatures promote growth through the phyB-PIF-auxin signaling module[5,11,12,25,26,30,35]. We asked what the response to both signals would be when presented simultaneously. Seedlings were grown in constant white light (WL) at 21 °C for 3 days. Afterward, they were kept in WL at 21 °C (21WL) or moved either to WL at 30 °C (30WL), to low R/FR at 21 °C (21FR), or to low R/FR at 30 °C (30FR, warm shade) for additional 3–5 days (Fig. 1a). We chose 30 °C for our experimental set-up since we found that maximal hypocotyl elongation in response to elevated temperature occurred between 28 and 30 °C (Supplementary Fig. 1a).

In agreement with previous reports, 21FR and 30WL promoted similar elongation of the hypocotyl (Fig. 1b, c). However, the treatment with both signals simultaneously (30FR) resulted in a much longer hypocotyl than either stimulus alone. We detected this effect as early as 4 h after the transfer to 30FR (Fig. 1c). This synergistic elongation was reflected in epidermal cell elongation throughout the hypocotyl (Fig. 1d, e), and was detectable at 24 °C, with maximum impact at 28–30 °C (Fig. 1f). Since the interaction between temperature and low R/FR most likely occurs in many crops grown in high density, we wanted to confirm that the synergistic growth response is conserved in crop plants. We found that the synergistic response also occurs in tomato (Solanum lycopersicon) and Nicotiana benthamiana (Supplementary Fig. 1b–f), emphasizing the importance of identifying the factors and the mechanisms that drive this interaction.

We next asked if the synergistic growth response observed at 30FR requires simultaneous light and temperature signals or if it can be achieved by consecutive treatments. We transferred seedlings between 21FR and 30WL, and between 30WL and 21FR at different time points and measured hypocotyl lengths. Only the seedlings that were exposed to simultaneous low R/FR and high temperature showed the synergistic elongation response (Fig. 1g, Supplementary Fig. 2a, b). Sequential presentation of these stimuli failed to enhance growth. We, therefore, conclude that simultaneous sensing of low R/FR and warm temperature is required for this enhanced growth response. Additionally, we noticed that in low light conditions (30 μmol m$^{-2}$ s$^{-1}$), the synergistic interaction between low R/FR and the warm temperature was suppressed (Supplementary Fig. 3a), indicating that this response is light intensity dependent.

### PIF7 is necessary and sufficient for the synergistic growth response

The importance and dominance of PIF4, and to some extent PIF5, in thermomorphogenesis has been shown in many studies[15,16,35,36]. Recently, PIF7, which has a dominant role in response to low R/FR, was also found to have a role in the thermomorphogenesis response[19,20,23,25]. We, therefore, asked which PIFs are essential for the synergistic response of elevated temperatures under low R/FR conditions. We used a genetic approach and examined this response in multiple pif mutants. In agreement with published data, we found that PIF7 and PIF4 played dominant roles in response to low R/FR and to warm ambient temperature, respectively (Fig. 2a). The pif7 mutant showed reduced hypocotyl elongation in warm temperatures, but this phenotype was not as strong as observed in pif4 or pif45 mutants. Surprisingly, we found that pif4 or pif45 mutants resembled wild-type seedlings in 30FR, while pif7 eliminated the synergistic elongation. This phenotype was slightly enhanced in the pif457 mutant (Fig. 2a). These results suggest that PIF7 has the dominant function when the elevated temperature is sensed together with low R/FR (30FR), while PIF4 and PIF5 are minor contributors. This pattern was also apparent under long-day conditions at 28 °C (Supplementary Fig. 3b). Interestingly, while pif457 suppressed the long hypocotyl phenotype of phyB in 21WL or 21FR, it only partially suppressed the phyB long hypocotyl phenotype at 30WL (Supplementary Fig. 3c, d), indicating that part of the response to 30WL in phyB plants is PIF457-independent, as shown before[37].

The activity of PIF7 in thermomorphogenesis is dependent on PIF4 and PIF5[19]. We asked if this relationship persisted in response to 21FR or 30FR. We tested a transgenic line expressing PIF7 driven by its native promoter (PIF7:PIF7-4xMYC) in the pif457 background[23]. This line failed to rescue elongation in 30WL but fully rescued elongation in 21FR or 30FR, perfectly phenocopying the pif45 mutant phenotype (Fig. 2a, b). The direct comparison between pif457 and pif45 suggests that PIF7 activity depends on PIF4 in 30WL, but does not require PIF4 or PIF5 in 21FR or for the exaggerated growth response to 30FR. Also evident by the complementation of pif457 by PIF7:PIF7-4xMYC in 21FR and 30FR conditions but not in 30WL. In addition, we found that PIF4 and PIF5 transcripts were upregulated in 30WL and 30FR while PIF7 expression did not respond to changes in growth conditions, suggesting different post-transcriptional mechanisms governing PIF4, PIF5, and PIF7 activity in 30WL and 30FR conditions (Supplementary Fig. 4a).

Next, we asked if the dominant function of PIF7 and the minor role of PIF4 observed under combined temperature and light changes are unique to low R/FR conditions. To test this, we chose Low Blue Light (LBL) since growth under LBL was shown to be regulated mainly by PIF4[38,39]. We combined LBL with warm ambient temperature, finding that while LBL and warm temperature interacted synergistically, PIF4 played the major role, while PIF7's contribution was smaller (Fig. 3a). Consistent with these results, and similar to 30WL, the PIF7:PIF7-4xMYC construct failed to fully rescue the elongation of pif457 mutant in LBL at 21 °C or 29 °C (Fig. 3a). We hypothesized that the functions of PIF7 in

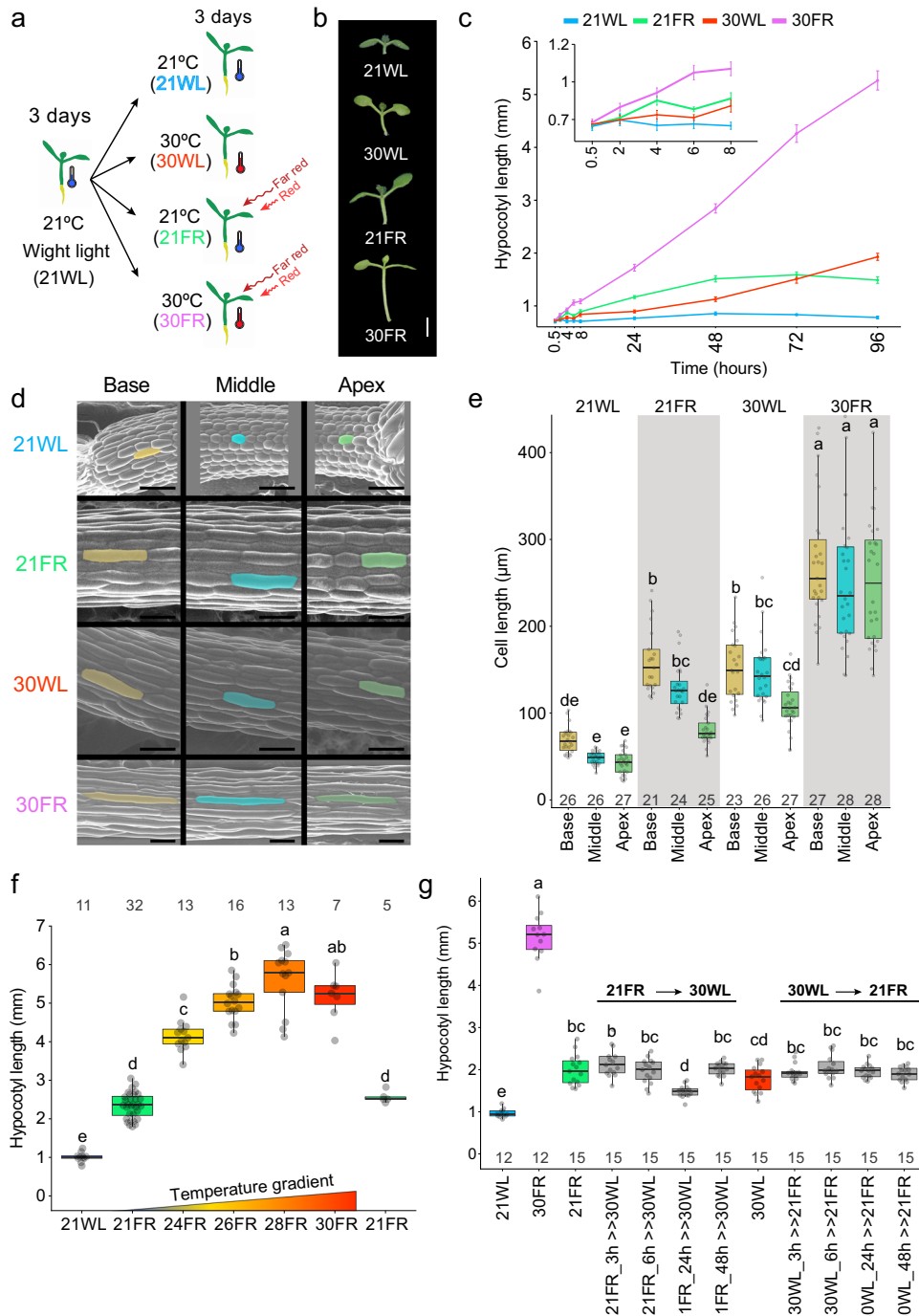

**Fig. 1 | Synergistic hypocotyl elongation under combined shade and warm temperature. a** Scheme of the experimental design. Seedlings were grown in 21 °C with constant simulated white light (WL) for 3 days and then moved to 30 °C (30WL), 21 °C with shade (21FR, R/FR = 0.6), 30 °C with shade (30FR, R/FR = 0.6) or kept at 21 °C white light (21WL) for three more days. **b** Representative images of 6-day-old Arabidopsis wild-type seedlings grown as described in (**a**). Scale bar = 2 mm. **c** Hypocotyl length of wild-type seedlings during the first 4 days after transfer to the different conditions. Insert is an enlargement of the first hours. The average values +/− SE are shown, n = 8 or more seedlings per time point (for the exact number in each sample, see Source data). **d** Representative SEM images of hypocotyl epidermal cell lengths at the apex, base, and middle of 6-day-old seedlings grown as described in (**a**). We repeated the experiment five times with similar results. Scale bar = 100 μm **e**. Hypocotyl epidermal cell length was determined from confocal images of 8–9 seedlings per condition and 2–3 cells per segment. The

number of cells (n) is shown under each box. **f** Hypocotyl length of 6-day-old wild-type seedlings grown for 3 days at 21WL and then moved to the indicated temperature and light. The response to 21FR is shown for two different chambers used for the experiment. The number of seedlings (n) is shown on top of each box. **g** Hypocotyl length of 6-day-old wild-type seedlings grown for 3 days at 21WL, then moved to the indicated conditions. In gray, the plates were transferred from 21FR to 30WL or from 30WL to 21FR, after the indicated time and stayed there for the rest of the experiment. The number of seedlings (n) is shown under each box. In **b**–**g**, PAR = 70 μmol m⁻² s⁻¹. In **e**–**g**, Different letters denote statistical differences (p < 0.05) among samples as assessed by one-way ANOVA and Tukey HSD. Boxes indicate the first and third quartiles and the whiskers indicate the minimum and maximum values, the black lines within the boxes indicate the median values and gray dots mark the individual measurements.

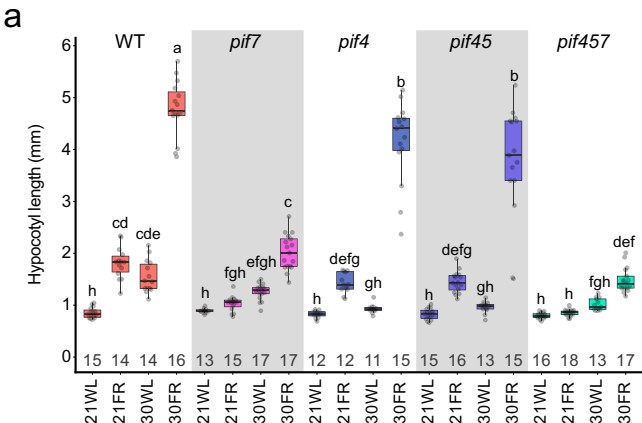

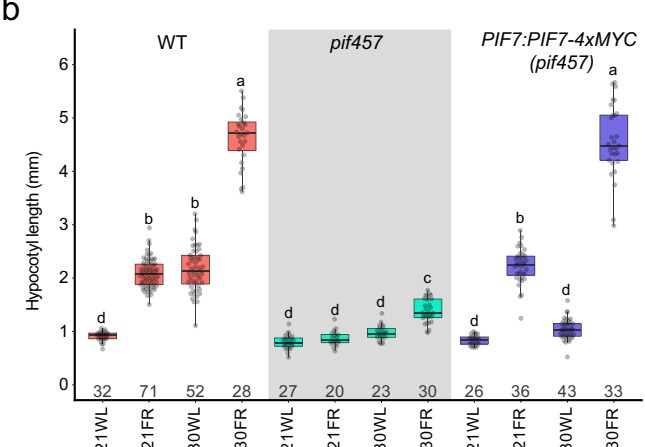

**Fig. 2 | PIF7 plays a dominant role in the synergistic response to low R/FR and warm ambient temperature. a** Hypocotyl length of 6-day-old Arabidopsis pif mutant seedlings. The number of seedlings (*n*) is shown under each box. **b** Hypocotyl length of 6-day-old Arabidopsis *PIF7:PIF7-4xMYC* (*pif457*) seedlings. The number of seedlings (*n*) is shown under each box. The growth conditions are the same as described in Fig. 1a, PAR = 70 μmol m$^{-2}$ s$^{-1}$. Different letters denote statistical differences ($p < 0.05$) among samples as assessed by one-way ANOVA and Tukey HSD test. Boxes indicate the first and third quartiles and the whiskers indicate the minimum and maximum values, the black lines within the boxes indicate the median values and gray dots mark the individual measurements.

low R/FR and PIF4 in LBL and thermomorphogenesis might result from their interactions with phyB and cry1. PIF7 being potentially regulated by phyB alone, while PIF4 is controlled by phyB and cry1[38–41]. To test this hypothesis, we compared hypocotyl elongation of *cry1* and *pif* mutants in response to 21FR, 30WL, and 30FR. We found that *pif7* suppressed the long hypocotyl phenotype of *cry1* in all tested conditions (Fig. 3b). This indicates that PIF7 works downstream of both phyB and cry1. We confirmed our finding in high monochromatic blue light (Supplementary Fig. 4b)[42].

All together these results demonstrated that PIF7 has a dominant role in response to elevated temperature when combined with low R/FR. Surprisingly, PIF4 has a minor role in the combined response to high temperature and low R/FR, while maintaining a major role in the combined response to warm temperature and LBL. This indicated separate mechanisms for processing warm temperatures depending on light quality.

**The response to 30FR condition is independent of phyB thermal reversion**
The phyB-PIF-auxin signaling pathway is central to growth responses in low R/FR and warm temperatures. PIF7 is regulated mainly by phyB[23,41],

and we found that PIF7 is the major PIF required for the synergistic response of low R/FR and warm ambient temperature. Therefore, we examined the contribution of phyB to this response and asked whether changes in PIF7 activity can explain the synergistic response to 30FR. We found that the relative response to 30WL in *phyB* mutants was similar to wild-type seedlings (Supplementary Fig. 3c, d). In addition, phyB abundance was similar in all growth conditions (Supplementary Fig. 3e, f). However, relative elongation growth in response to low R/FR at either 21 °C or 30 °C was compromised in *phyB* mutants (Fig. 3c, Supplementary Fig. 3c, d). As the phyB thermal reversion rate was shown to increase with increasing ambient temperature[11,12], we asked if this phenomenon might contribute to the synergistic response to 30FR. We tested transgenic lines that overexpress PHYB with a modified S86 phosphorylation site leading to faster reversion (phyB$^{S86D}$-YFP) or slower reversion (phyB$^{S86A}$-YFP)[43]. Phosphorylation of this serine residue varies based on low R/FR and temperature conditions[44]. Plants over-expressing phyB-GFP or phyB$^{S86A}$-YFP had shorter hypocotyls than controls in all tested conditions (Fig. 3c). However, overexpression of phyB$^{S86D}$-YFP, led to a wild-type-like hypocotyl phenotype in response to 21FR and 30FR but failed to elongate in 30WL (Fig. 3c). These results suggest that in our light conditions (continuous white light), the effect of increased temperature on phyB activity is minor relative to the impact of low R/FR light. In addition, the synergistic effect seen in response to 30FR cannot be explained by decreased phyB activity under simultaneous low R/FR and warm temperature.

Next, we asked whether PIF7 activity increases in 30FR. Since PIF7 is regulated by phosphorylation, we tested whether there is a difference in PIF7 phosphorylation between 21FR and 30FR. We found that PIF7 was dephosphorylated as early as 10 min in the low R/FR and stayed dephosphorylated as long as it remained in low R/FR in both temperatures (Fig. 4a, Supplementary Fig. 5). PIF7 was rapidly phosphorylated when transferred from 21FR to 30WL (Fig. 4b). This result suggests that PIF7 is dephosphorylated similarly by low R/FR in 21 °C and 30 °C. It was recently shown that PIF7 protein level increases during the thermomorphogenesis response[19,20]. We did not observe difference in PIF7 levels when seedlings were transferred from 21WL to 21FR versus 30FR (Fig. 4a, Supplementary Fig. 5). We, therefore, asked whether PIF7 is more active in promoting the transcription of its targets in 30FR compared to 21FR or 30WL. We performed a global transcriptomic analysis using RNA-sequencing (RNA-seq) and chromatin immunoprecipitation-sequencing (ChIP-seq) and compared the expression levels of direct PIF7 targets under 21WL, 21FR, 30WL, or 30FR. The majority of PIF7 direct targets were upregulated in both 21FR and 30FR (Fig. 4c–f). In agreement with this result, PIF7 binding to the DNA was similar in 21FR and 30FR (Fig. 4g, h, Supplementary Data 4). These results suggest that PIF7 has the same activity under 21FR and 30FR conditions. Interestingly, in the 30WL condition, PIF7 binds to very similar sets of genes as in 21FR and 30FR conditions, while almost no binding was observed in 21WL (Fig. 4g, h, Supplementary Data 4). In addition, PIF7 binding in the 30WL condition was insufficient to rescue the *pif457* mutant response or promote the expression of target genes such as *IAA29* and *ATHB2* (Figs. 2b, 4d, e, h). These results indicate that while warm temperature promotes the binding of PIF7 to its targets, its activity depends on the interaction with additional co-factors.

We concluded that deactivation of phyB by low R/FR, dephosphorylation of PIF7, and activation of PIF7-mediated transcription are necessary for the enhanced elongation at 30FR. However, these steps cannot explain the synergistic effect observed in 30FR since we observed these low R/FR responses independent of the ambient temperature.

**Auxin-dependent hypocotyl elongation differs in response to 21FR, 30WL, and 30FR**
Both low R/FR and warm temperature promote growth through the canonical phyB-PIF signaling pathway, which controls the levels of

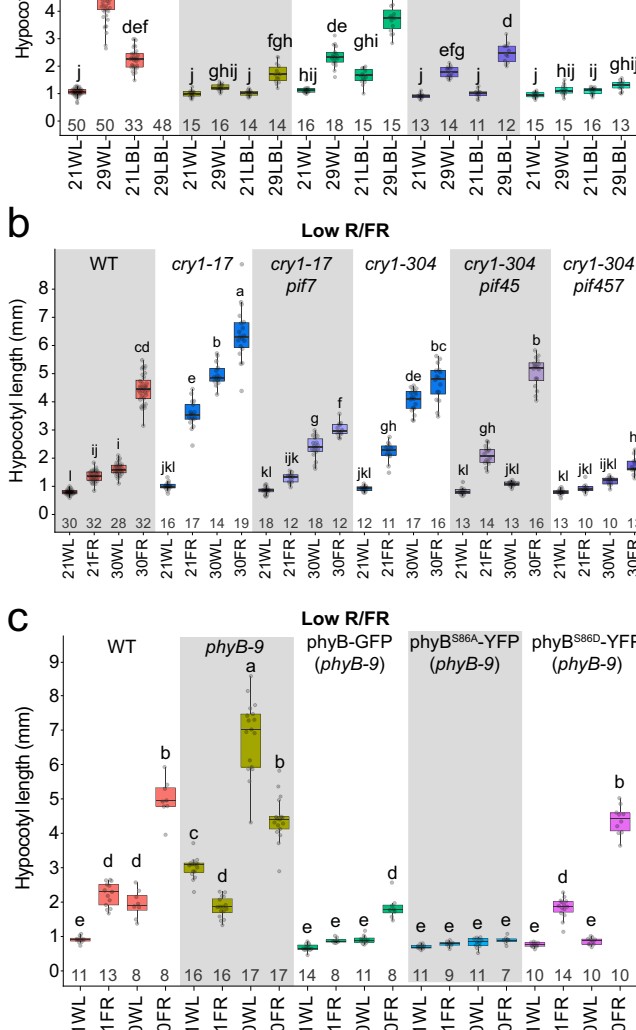

**Fig. 3 | PIF7 acts downstream of both *PHYB* and *CRY1*. a** Hypocotyl length of 6-day-old wild-type Arabidopsis seedlings grown for 3 days at 21 °C, then moved to the indicated conditions. Long-day white light (WL, fluorescent bulbs, PAR = 60 μmol m$^{-2}$ s$^{-1}$), or Low Blue Light (LBL, PAR = 56 μmol m$^{-2}$ s$^{-1}$, Supplementary Data 1). Temperature: constant 21 °C or 29 °C. The number of seedlings (*n*) is shown under each box. **b** Hypocotyl length of 6-day-old Arabidopsis *cry1* and *pif* mutant seedlings. The number of seedlings (*n*) is shown under each box. **c** Hypocotyl length of 6-day-old *phyB* seedlings expressing *35S:PHYB-GFP*, *35S:PHYB$^{Ser86Asp}$-YFP* (S86A), or *35S:PHYB$^{Ser86Ala}$-YFP* (S86D). The number of seedlings (*n*) is shown under each box. In **b**, **c**, growth condition is the same as described in Fig. 1a, PAR = 70 μmol m$^{-2}$ s$^{-1}$. Different letters denote statistical differences (*p* < 0.05) among samples as assessed by one-way ANOVA and Tukey HSD test. Boxes indicate the first and third quartiles and the whiskers indicate the minimum and maximum values, the black lines within the boxes indicate the median values and gray dots mark the individual measurements.

auxin[19,25,35,45–47]. Therefore, control of the auxin levels and auxin sensitivity are appealing candidates to explain the synergistic growth response at 30FR. To test this, we determined the levels of auxin (indole-3-acetic acid, IAA) and IAA metabolites under our growth

conditions. We observed a slight increase in IAA levels in response to 21FR and 30FR after 3 h. In addition, we observed a substantial increase in the levels of several IAA conjugates. IAA-Glutamate (IAA-Glu) accumulated in response to 30FR and 21FR, while IAA-Aspartate (IAA-Asp) increased only in response to 30FR. The changes in IAA-Glu and IAA-Asp levels correlated well with the expression of the *GH3* genes, which regulate their accumulation (Supplementary Fig. 6a, b)[48,49]. In response to 30WL, we only observed a change in the IAA-glucose (IAA-glc) level. However, we did not detect changes in the expression levels of the enzymes that generate this conjugate: *UGT84B1* (below detection) and *UGT74D1*[50–52] (Supplementary Fig. 6a, Supplementary Data 2). The accumulation of IAA conjugates suggests an increase in IAA after exposure to 21FR, 30WL, or 30FR, most likely prior to the time point of sampling. Next to test the contribution of auxin and its transport to hypocotyl elongation under these conditions, we applied NPA (*N*–1-naphthylphthalamic acid, an inhibitor of auxin transport) or peo-IAA (an inhibitor of auxin perception). We found that growth in all four conditions is dependent on auxin transport and auxin perception, as previously reported for low R/FR and thermomorphogenesis (Supplementary Fig. 6c)[27,53]. These results also indicate that the accumulation of IAA and its conjugates in response to low R/FR and to high temperature is highly dynamic. We also examined the role of the *YUCCA (YUC)* auxin biosynthetic enzymes. We used a high order *yucca* mutant, *yucca2589*, which is known to suppress the response to low R/FR[30]. The *yucca2589* mutant strongly suppressed the response to 21FR but only slightly suppressed the response to 30WL. The response to 30FR was similar to 30WL (Supplementary Fig. 6d). While *YUC2,5,8* and *9* were shown to be important for thermomorphogenesis under long-day conditions[19], our results suggest that in constant light, they are primarily involved in the low R/FR response, with only a minor role in the temperature response.

Given the minor differences observed in auxin biosynthesis among our conditions, we asked if the increased elongation under 30FR may be due to increased sensitivity to auxin. HSP90 proteins were shown to increase the sensitivity to auxin during the response to warm temperature by stabilizing the TIR1 auxin receptor[54]. In addition, in our transcriptomic analysis, we found that many HEAT SHOCK PROTEINS (HSPs) and HEAT SHOCK FACTORS (HSFs) were upregulated in 30WL and 30FR (Supplementary Fig. 7a, Supplementary Data 3). We hypothesized that HSP90s and HSFs might explain the synergistic response at 30FR. To test this hypothesis, we blocked HSP90 activity using Geldanamycin (GDA) and, in agreement with published data, we observed dose-dependent inhibition of hypocotyl elongation in 30WL and 30FR (Supplementary Fig. 8a)[54]. Next, we decided to test the *SUPPRESSOR OF G2 ALLELE SKP1b* (*SGT1b/eta3*), which is crucial for HSP90 activity[55]. In contrast to our results with GDA treatment, the *eta3* mutant showed only a minor decrease in hypocotyl length at 30WL and 30FR and still responded in a synergistic manner to 30FR (Supplementary Fig. 8b). To address this discrepancy, we used the shade responsive *PIL1* promoter (*PIL1p*, Supplementary Fig. 8c) to conditionally overexpress *HSFA2*, HSP90.1, *HSP90.2*, and a dominant-negative mutant form HSP90.2(D80N), which binds but does not release its client proteins. We hypothesized that if HSFA2 and HSP90 promote growth in response to warm temperatures, shade-dependent overexpression will promote further hypocotyl growth in 21FR, while HSP90.2(D80N) will suppress growth in 30FR. However, while the expression of these proteins was induced by low R/FR, we could not detect enhancement of growth by HSFA2, HSP90.1 or HSP90.2, or inhibition of growth by HSP90.2(D80N). All lines behaved like the wild-type control under all conditions (Supplementary Fig. 8d, e). Therefore, we cannot confirm or reject the hypothesis that HSP90s are responsible for the synergistic response to low R/FR and warm temperature.

If simultaneous low R/FR and warm temperature specifically modify auxin sensitivity, an increase in the expression of auxin response genes could be expected. To test whether the auxin response

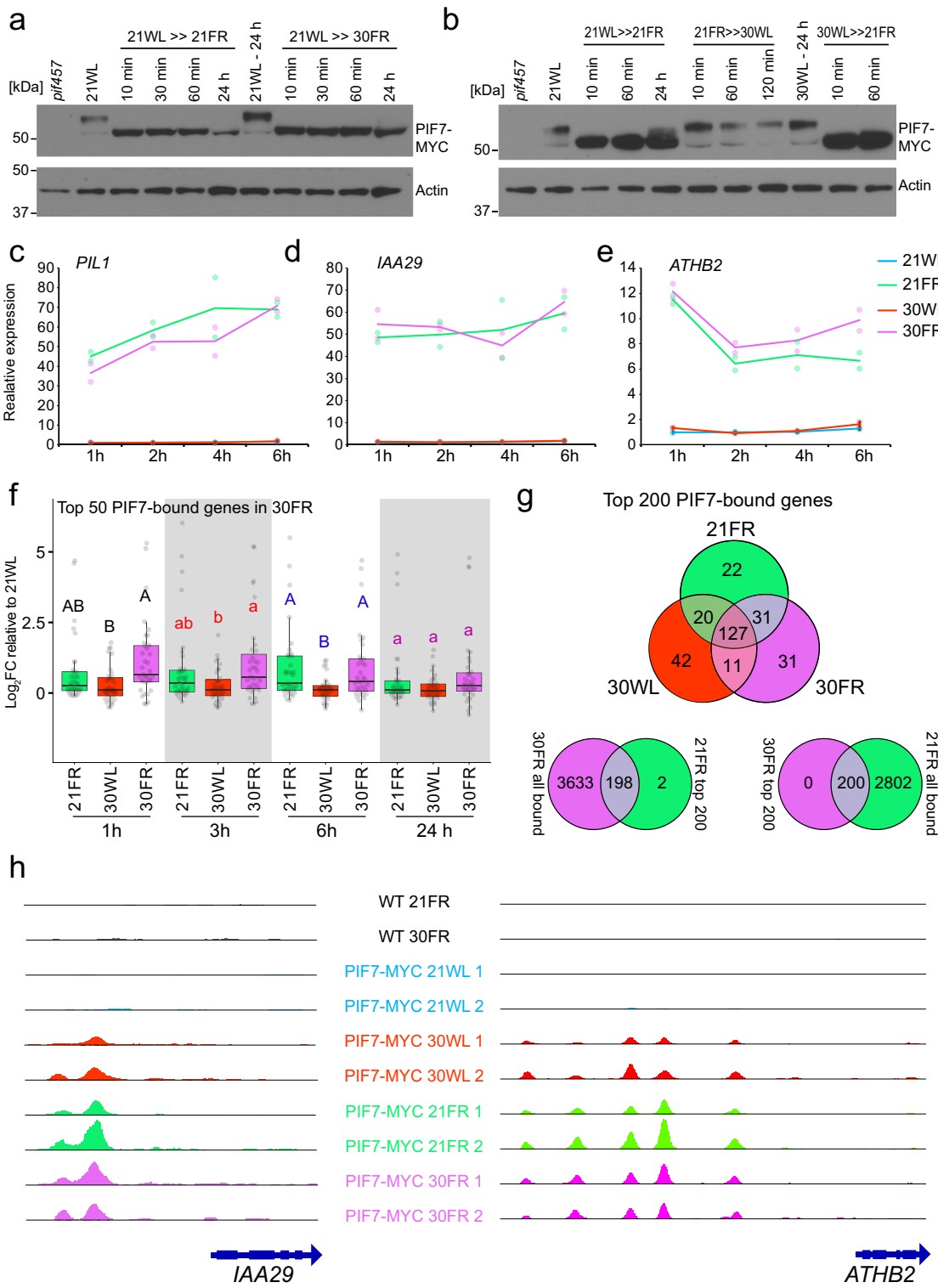

differs between 21FR, 30WL, and 30FR, we compared the expression of *IAA29, YUC8, YUC9, SMALL AUXIN UP RNA 19 (SAUR19)*, and *SAUR22*, all of which are upregulated in response to auxin, low R/FR, or warm temperature[25–27,30,32,35,39,56,57]. In 21FR and 30FR, RT-qPCR showed that the expression levels of *IAA29, SAUR19, SAUR22, YUC8*, and *YUC9* were massively increased (6–70-fold over 21WL), while in response to 30WL, their expression was only slightly increased (1.5–3-fold over 21WL) (Figs. 4d, 5a). We observed similar patterns in the whole transcriptome data (see cluster 3 in Supplementary Fig. 7a and Supplementary Data 3), as well as a narrower comparison of *SAUR* genes (Fig. 5b). In

addition, examination of *SAUR19p:GUS* and *DR5:GUS* reporters lines showed that these changes occur in cotyledons and in hypocotyls (Fig. 5c and Supplementary Fig. 6e). In agreement with the finding that *PIF7* has a dominant role in response to 21FR and 30FR, we found that the expression of *YUC8* in these conditions was well correlated with *PIF7* activity (Fig. 5d). *SAUR22* expression, on the other hand, was dependent on *PIF4, 5*, and *7*, but its expression levels did not tightly correlate with hypocotyl elongation: We observed no difference in expression between the *pif7* mutant and wild-type controls in 21FR or 30FR conditions, although *pif7* hypocotyls in these conditions are

**Fig. 4 | PIF7 activity is comparable in low R/FR at low or high temperature (21FR vs 30FR conditions). a** Immuno-detection of PIF7-MYC Protein levels using anti-MYC antibody. Total protein extract was collected from 3-day-old *PIF7:PIF7-4xMYC (pif457)* seedlings grown in 21WL (70 μmol m$^{-2}$ s$^{-1}$) plus the indicated time in the stated conditions. Anti-ACTIN blots are shown below as loading controls. This experiment was done one time. **b** Same as (**a**) except protein extract was collected after 24 h in the indicated starting condition plus the stated time in the treatment condition. This experiment was done one time. **c**–**e** Expression of *PIL1* (**c**), *IAA29* (**d**), and *ATHB2* (**e**) in 3-day-old seedlings grown at 21 °C in 21WL (70 μmol m$^{-2}$ s$^{-1}$) plus the indicated time in the stated conditions. The expression from the RNA-seq data is shown as normalized counts per million reads mapped (cpm), and as the average of two biological replicates per condition. Color dots mark the individual measurements. **f** Relative expression of the top 50 PIF7-bound genes in 30FR that are also differentially expressed in at least one condition of the RNA-seq (FDR < 0.05, *n* = 31). Log$_2$FC: Log$_2$ Fold Change relative to 21WL at each time point. Different letters denote statistical differences (*p* < 0.05) within time points assessed by one-way ANOVA and Tukey HSD. Boxes indicate the first and third quartiles and the whiskers indicate the minimum and maximum values, the black lines within the boxes indicate the median values and gray dots mark the individual measurements. **g** Venn diagram comparing the top 200 PIF7-bound genes in 21FR, 30WL, and 30FR. Bound genes were annotated if a peak was found 2 kb upstream to the TSS or in the gene body in both replicates of each growth condition. All top 200 genes in 30FR were also bound in 21FR (all genes) and all top 200 genes in 21FR were also bound in 30FR (all genes) except two (see Supplementary Data 4 for the top 200 genes in each condition). **h** Visualization of PIF7-MYC binding to the *IAA29* and *ATHB2* promoters. Note that differences in binding peak intensities between replicates might be due to differences in library preparation and data analysis between replicates 1 and 2 (see "Methods" section). Light and temperature conditions in **a**–**g** were the same as in Fig. 1.

significantly shorter than wild-type hypocotyls (Figs. 2a, 5d). Overall, these results suggest that the magnitude of the auxin response differs between 21WL and 30WL but cannot explain the increased growth in 30FR compared to 21FR.

Our data set revealed few genes that were differentially expressed only in the 30FR condition during the first 6 h. Upon closer analysis, we found that not a single gene was differently expressed across all time points (Supplementary Fig. 7b, c, e and Supplementary Data 3). This suggests that there are no unique gene sets differentially expressed in response to 30FR. In agreement with this, most of these genes were not bound by PIF7 (Supplementary Fig. 7d). However, the dynamics of expression of a relatively small number of genes that are differently regulated in 30FR, such as *EXP8* and *IAA19*, did vary between conditions (Supplementary Fig. 7f).

### Warm temperature enhances auxin-mediated hypocotyl growth

The minor changes observed in auxin response gene expression, auxin synthesis, and auxin signaling between 30FR and 21FR led us to hypothesize that increased temperature might increase the efficiency of auxin in promoting growth. To test this, we treated seedlings with the auxin analog picloram and measured hypocotyls after 3 days on the drug (treatment starting day 2: Fig. 6a, b, or day 3: Supplementary Fig. 9a). Without picloram, hypocotyls at 30WL were nearly identical to those at 21FR. However, we observed that elongation growth in response to increasing picloram concentration was stronger at 30WL than at 21WL or 21FR. Even at the lowest concentrations, picloram caused an increase in hypocotyl length in 30WL and mimicked those of untreated seedlings at 30FR. We did not observe a response of this magnitude at 21WL or 21FR. These treatments suggest that high temperatures enhances auxin-mediated hypocotyl growth (Fig. 6a, b).

Together, our results indicate that the role of auxin in thermomorphogenesis is different from its role in low R/FR. While 21FR caused a dramatic increase in auxin response, 30WL led only to a moderate increase. However, both signals led to similar hypocotyl elongation phenotypes. Therefore, our data indicate that while auxin is necessary to initiate thermomorphogenesis, growth at 30WL depends on additional unknown factors that increase the efficiency of auxin in promoting elongation. Hence, in seedlings that sensed low R/FR and elevated temperature simultaneously, low R/FR might induce auxin production while warm temperature might induce the expression or the activity of other factors that likely work downstream to auxin and increasing the effects of auxin on hypocotyl growth (Fig. 6c). The interaction between the high level of auxin response and increase in auxin efficiency in regulating growth, results in the observed hyper-elongated phenotype in 30FR.

## Discussion

Plants continuously monitor the surrounding environment and tune their growth and developmental programs accordingly. In response to a non-optimal condition, such as an increase in the ambient temperature or change in the light quality, a sign for nearby vegetation, plants will promote stem elongation and flower early[3,5,56,58]. While these environmental conditions are usually studied separately, plants often sense them simultaneously, for instance, during the warm season[10]. We examined responses to low R/FR and warm ambient temperature separately and simultaneously, finding that these signals caused a dramatic synergistic effect on hypocotyl elongation. We identified PIF7 as the dominant transcription factor which regulates this synergistic growth. We showed that the auxin induction during neighbor detection is significantly higher than during the response to warm ambient temperature (Fig. 5), suggesting that auxin alone is not sufficient to explain the full response to warm temperature. We propose that another factor must be active during thermomorphogenesis, which boosts the efficiency of auxin in promoting hypocotyl elongation. Our results suggest that this factor works in parallel or downstream to the photoreceptors-PIF-auxin module and is likely to explain the synergistic interaction between warm temperature and low R/FR responses, and to some extent with LBL light.

We show that *PIF7* plays a dominant role in the synergistic interaction between low R/FR and warm temperature, similar to its function in low R/FR (Fig. 2a). *PIF4*, and to some extent *PIF5*, were shown to be the major *PIF* family members controlling hypocotyl elongation during thermomorphogenesis[5,59]. Surprisingly, when the warm ambient temperature was sensed together with low R/FR, the roles of *PIF4* and *PIF5* were minor (Fig. 2a).

In two recent papers, *PIF7* was shown to regulate the warm temperature response along with *PIF4*[19,20]. Those groups demonstrated that PIF7 accumulates at the protein level during this response. However, we found that the abundance and the activity of PIF7 did not change at 30WL or 30FR compared to 21WL or 21FR, respectively (Fig. 4a, b, Supplementary Fig. 5). As the published work was carried out in LD conditions, we suggest that the accumulation of PIF7 protein is dependent on cycling conditions. We observed the synergistic growth phenotype under both constant light and LD conditions. Therefore, we suggest that even if PIF7 protein accumulation differs under these conditions, it is likely not a determining factor in promoting growth under warm shade (30FR) condition.

As our data shows and as described previously, PIF7 activity to promote growth in elevated temperatures seem to be dependent on PIF4. To explain the phenotypes of both *pif4* and *pif7* during thermomorphogenesis, the authors suggested that the interaction between PIF4 and PIF7 proteins may be crucial[19]. Our finding that *pif45* has a minor effect on hypocotyl elongation and that *PIF7:PIF7-4xMYC* rescues the *pif457* growth defect at 21FR or 30FR but not at 30WL suggests that PIF7 activity depends on PIF4 only in response to warm temperature but not in response to low R/FR (at 21 °C or 30 °C) (Fig. 2b).

One possible explanation for this specificity could be that in low R/FR, the majority of PIF7 protein is released from the regulation by

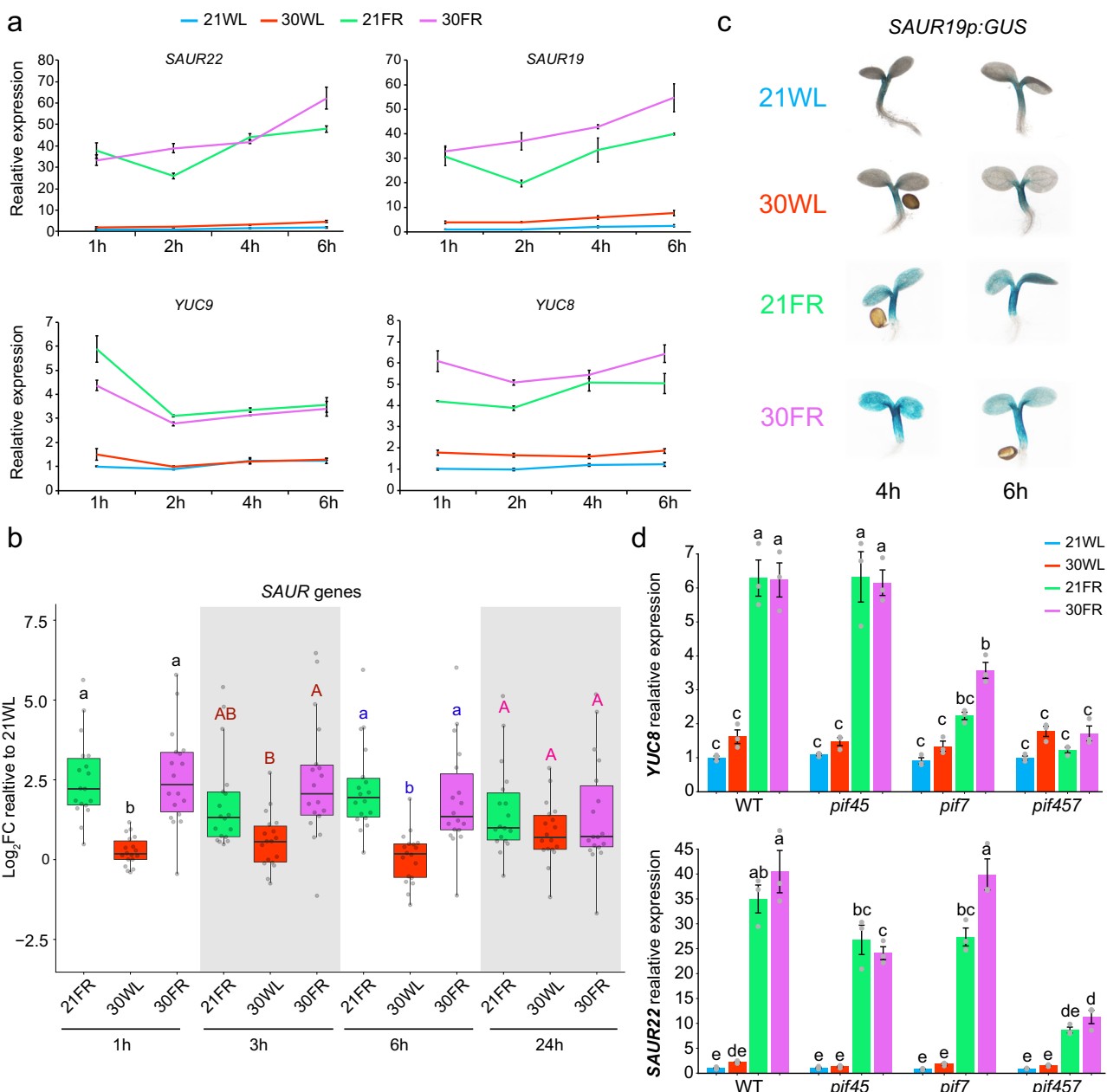

**Fig. 5 | Auxin response is higher in shaded conditions than in warm temperature and is predominantly regulated by PIF7 in shade at 21 or 30 °C.**
**a** Expression of *SAUR22*, *SAUR19*, *YUC9*, and *YUC8* in 3-d-old seedlings grown in constant white light (70 µmol m⁻² s⁻¹) at 21 °C plus the indicated time in the stated conditions. Relative expression assayed using RT-qPCR relative to the reference gene *IPP2*. The average values of three biological replicates per condition +/− SE shown. **b** Relative expression of all *SAUR* genes that show differential expression in at least one condition assayed by RNA-seq (Log₂FC < −1 or >1, FDR < 0.05). Expression in 3-day-old seedlings (grown in constant simulated white light 70 µmol m⁻² s⁻¹ at 21 °C) plus the indicated time in each of the conditions, shown as Log₂FC relative to 21WL. Different letters denote statistical differences (*p* < 0.05) between conditions, within each time point as determined by one-way ANOVA and Tukey HSD test. Boxes indicate the first and third quartiles and the whiskers indicate the minimum and maximum values, the black lines within the boxes indicate the median values and gray dots mark the individual measurements *n* = 18 genes. **c** Images of GUS-stained whole Arabidopsis seedlings carrying a *SAUR19p:-GUS* reporter. Plants were grown in constant simulated white light (70 µmol m⁻² s⁻¹) at 21 °C plus 6 h or 8 h in the conditions indicated on the left. **d** Expression of *YUC8* and *SAUR22* in 3-d-old wild-type, *pif45*, *pif7*, and *pif457* seedlings grown in constant simulated white light (70 µmol m⁻² s⁻¹) at 21 °C plus 4 h in the stated condition. Gene expression was assayed using RT-qPCR relative to the reference gene *IPP2* and normalized to the expression in wild type at 21WL. The average values of three biological replicates per condition +/− SE are shown. Different letters denote statistical differences (*p* < 0.05) among samples as assessed by one-way ANOVA and Tukey HSD test.

phyB, while PIF4 and PIF5 are still tightly regulated by cry1[25,38,39,60]. This could also explain the mild phenotypes of *pif45* in low R/FR conditions (Fig. 2a, Supplementary Fig. 3b). However, at warm temperatures, there are still pools of active phyB and cry1, and as such, only a few molecules of PIF7 and PIF4 might escape from photoreceptor-mediated repression. As a result, any small change in the levels of

PIF7 or PIF4 will have a dramatic effect on this response. As the PIF4 transcript level accumulates in response to a warm temperature (Supplementary Fig. 4a), changing PIF4/PIF7 ratio may modify the relative binding of phyB to the PIFs due to different affinities, for example, if PIF7 interaction with phyB sequesters phyB from PIF4, leaving more PIF4 protein to bind DNA. The relatively small changes in

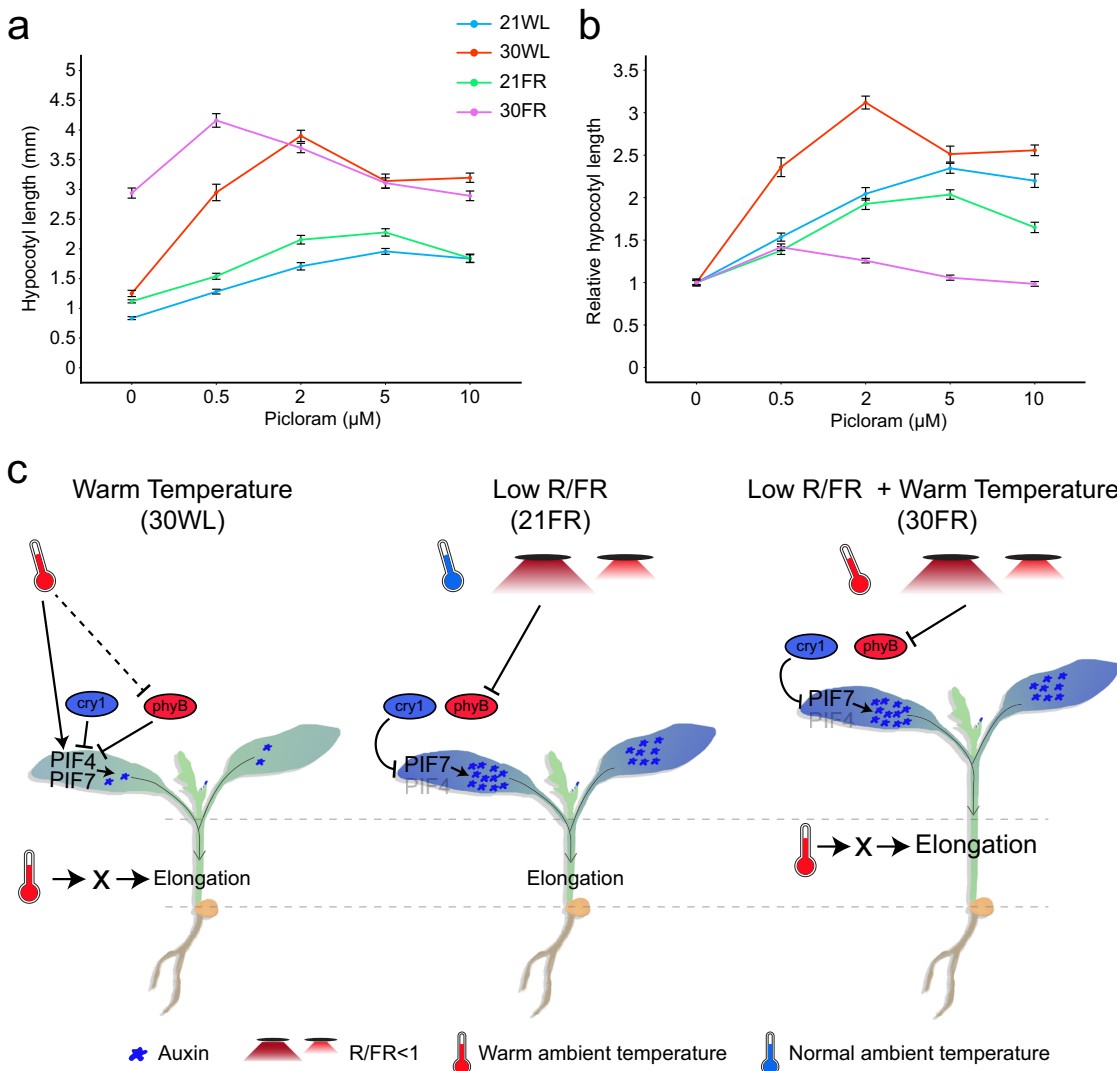

Fig. 6 | **Warm temperature enhances the response to auxin. a** Dose–response curves for hypocotyl length of 5-day-old wild-type Arabidopsis seedlings treated with the synthetic auxin picloram. Seedlings were grown in constant simulated white light (70 μmol m$^{-2}$ s$^{-1}$) at 21 °C (21WL) for 2 days, then moved to plates with the indicated picloram concentration or DMSO control (0) and the indicated growth condition for an additional 3 days. Data represent mean +/− SE; $n = 16$ seedlings or more per sample (for the exact number in each sample, see Source data). **b** Hypocotyl length in response to picloram shown in (**a**), normalized to the hypocotyl length of mock treatments in each growth condition. Data represent mean +/− SE; $n = 16$ seedlings or more per sample (for the exact number in each sample, see Source data). **c** A model of hypocotyl elongation in response to low R/ FR, warm temperature, and both signals simultaneously. During warm temperatures and high R/FR (30WL), the abundance of PIF4 and to some extent PIF7 increases, while both PIFs are still repressed by cry1 and phyB. This leads to a slight increase in auxin levels which initiate the growth response but depends on additional unknown factors (X). During the response to low R/FR (21FR), the repression of phyB over PIF7 is removed and auxin is induced, which is the main driving force that promotes hypocotyl elongation. In this condition, PIF4 plays a minor role (represented by faint font). When plants sense low R/FR and warm temperature simultaneously (30FR), auxin production increases as in the low R/FR, while the warm temperature activates factor X, boosting the effect of auxin in the hypocotyl resulting in extensive growth. The dashed line represents the increased dark reversion rate of phyB in cycling conditions.

the expression of auxin response genes, such as *YUC8* and *SAURs*, during thermomorphogenesis support this hypothesis. Our finding that in 30WL, PIF7 binds to target promoters but does not activate gene expression suggests that PIF7 requires other factors to promote transcription. It is possible that PIF4-PIF7 heterodimerization is important in this scenario.

We demonstrated a general interaction between warm temperature and light quality. Warm temperature synergistically interacts with LBL, consistent with previous studies finding that high blue light suppresses the elongation response to warm temperature, mainly through cry1[39]. Interestingly, this interaction is regulated not only by PIF4, a known interacting partner of cry1, but also by PIF7, which as we show also acts downstream to cry1 (Fig. 3b, Supplementary Fig. 4b). The

synergistic interactions between warm temperature and low R/FR or LBL are further evidence that warm temperature modulates hypocotyl growth downstream of the photoreceptor-PIF-auxin module. In line with this hypothesis, a previous study showed that LBL and low R/FR, which increase PIF activity, interact in an additive manner but not synergistically[61].

We propose that the specific activities of PIFs are determined by their availability, expression level, and interaction with the photoreceptors, in the case of warm temperature, also by their interaction with the other PIFs. This hypothesis will need to be addressed in future work.

Both the low R/FR and warm temperature responses have been suggested to promote elongation through the PIF-auxin

module[19,25,35,45–47,62]. Here we show that in some cases, these responses are fundamentally different. Both responses lead to the activation of PIFs, which in turn promote hypocotyl elongation through auxin. Warm temperature, only weakly induced expression of auxin biosynthetic and auxin response genes. A much stronger increase occurs in response to low R/FR. Even though the levels of the auxin response were significantly different between low R/FR and warm temperature, these signals promote hypocotyl elongation to a similar extent. This suggests that during the response to warm temperature, additional factors that are regulated by temperature and act downstream of the PIF-auxin module are involved. While during the response to 30FR, most of the growth regulated genes and auxin response genes were expressed as in 21FR, a few individual genes were expressed higher or upregulated earlier than in 21FR (Fig. 5, Supplementary Fig. 7). Therefore, we cannot exclude the possibility that a collection of small differences in gene expression and their expression dynamics contribute to the synergistic growth response. However, when taking into account the direct comparison between 21FR and 30WL, we consider this unlikely. The expression of auxin response genes at 21FR was significantly higher than in 30WL, but hypocotyl elongation was similar in both conditions. Therefore, we suggest that the synergistic response at warm low R/FR is the product of the low R/FR-induced activity of PIF7, which promotes a rapid increase in auxin response, combined with unknown warm temperature-induced factors which increase the effects of auxin on cell elongation (Fig. 6c). In support of this model, when seedlings were shifted from 21FR to 30WL or vice versa, they did not elongate as in 30FR, which correlated well with the rapid phosphorylation and dephosphorylation of PIF7. While this does not reveal the missing factors, it suggests that the activity of these factors is controlled rapidly and reversibly, similar to PIF7's phospho-status.

Our results indicate a temperature-sensitive factor or mechanism that works downstream of PIFs and auxin. The current model of auxin-mediated growth suggests that auxin promotes the expression of the SAUR proteins, which then inhibit the activity of type-2C protein phosphatases (PP2C.D). The inhibition of PP2C.D causes an increase in the activity of plasma membrane $H^+$-ATPases[63–65]. As a result, the concentration of protons in the apoplast increases, and the apoplast pH is reduced, leading to loosening of the cell wall and increased water uptake into the cells[66,67]. Induction of SAUR gene expression, regulation of phosphatase activity, and decrease of apoplastic pH are appealing candidates to fill the role of the missing factor. However, given the observation that PP2C accumulates in warm temperature and that overexpression of a stable form of SAUR19 (GFP-SAUR19) is not hypersensitive to low R/FR or warm temperature, suggest that these components are probably not the missing temperature-sensitive factor (Supplementary Fig. 9b)[35,68]. $H^+$-ATPases may be strong candidates for future work in this direction.

HSP90 proteins were shown to increase the sensitivity to auxin during the response to warm temperature; therefore, they could be the temperature-sensitive factor[54]. However, the mild phenotype of eta3 mutant and the lack of phenotype of HSP90 or HSFA2 overexpression lines suggest that this is not the primary mechanism of increasing auxin sensitivity, at least in our growth condition. The mild activation of auxin response genes at 30WL in comparison to 21FR supports this conclusion.

Another possibility is that the cell membrane fluidity or the cell wall rigidity and permeability may be regulated directly by temperature, potentially leading to exaggerated elongation at 30FR. Since our results show that the synergistic response can be reversed relatively fast (Supplementary Fig. 2b), the changes in the cell wall need to follow this pattern. Introducing new material into the cell wall is a relatively slow process. Therefore, it is more likely that the warm temperature induces growth by modifying the physical interactions between the existing cell wall components instead of by the formation of new cell wall material. For example, the increasing temperature might change

pH, ROS, or intermolecular interactions, resulting in modified cross-linking between cellulose and hemicellulose.

The scientific community has had several breakthroughs in the last decade in our understanding the growth related temperature responses, but these breakthroughs were primarily focused on the regulation of the phyB-PIF-auxin module. Here, we present evidence for another layer or regulation downstream of the PIF-auxin module that may be just as important for processing multiplexed environmental signals. In addition, we demonstrated that the interaction of low R/FR with warm ambient temperature is conserved in various plant species. In the context of the predicted increase in the average temperature, these studies begin to address mechanisms of potential yield loss due to the amplification of the low R/FR effect on yield by warm temperature. Therefore, a better understanding of this interaction and identifying the key factors that regulate it will aid the effort to breed better-performing crops for planting in high density. In this context, our finding that in *Arabidopsis thaliana* PIF7 is necessary and sufficient to mediate this response will facilitate our effort to achieve this goal.

## Methods

### Genetic material

All Arabidopsis (*Arabidopsis thaliana*) plants used were in the Col-0 background. The *pif4-2, pif4-2 pif5-3 (pif45)*[69], *pif7-1*[41], *pif4-2 pif5-3 pif7-1 (pif457)*[23], *phyB-9*[13], *phyB-9 pif4-101*[69], *phyB-9 pif7-1*[41], *phyB-9 pif4-101 pif5-2 (phyBpif45), phyB-9 pif4-101 pif5-2 pif7-1 (phyB pif457)*[23], *cry1-17*[70], *cry1-304*[71], *cry1-304 pif4-101 pif5-3 (cry1pif45), cry1-304 pif4-101 pif5-3 pif7-1 (cry1pif457), phyB-9 cry1-304*[19], *yuc2yuc5yuc8yuc9*[30], *eta3*[72], *PIF7:-PIF7-4xMYC*[23], *SAUR19p:GFP-GUS*[29], *35S:GFP-SAUR19*[57], *35S:phyB-GFP, 35S:phyB^{Ser86Asp}-YFP,* or *35S:phyB^{Ser86Ala}-YFP*[43] and *DR5:GUS*[73] were described previously. *cry1-17 pif7-1 was* generated by crossing and confirmed by genotyping *pif7-1* and sequencing *cry1-17* (primers listed in Supplementary Table 1).

### DNA constructs and plant transformation

The multisite Gateway system (Invitrogen) was used. To generate the *PIL1* promoter (*PIL1p*) a 1654-bp fragment upstream of the *PIL1* start codon was amplified by PCR from Col-0 genomic DNA and cloned into pDONR221 P4-P1r. To generate the *HSP90.1 (AT5G52640), HSP90.2 (AT5G56030),* and *HSFA2 (AT2G26150)* constructs, full-length coding sequence (without a stop codon) was PCR-amplified from Col-0 cDNA (*HSP90.1* and *HSP90.2*) or genomic DNA (*HSFA2*) and recombined into pDONR221 P1-P2. To generate the *HSP90.2(D80N)* in pDONR221 P1-P2 plasmid, a point-mutation (G → A) was introduced in *HSP90.2* by PCR, and the template was digested with DpnI prior to transforming *E. coli*. All genes were inserted downstream of *PIL1p* in pDONR221 P4-P1r and upstream to 4XMYC tag in pDONR221 P2r-P3 by Gateway recombination into the destination vector pH7m34GW. *PIL1p:GUS-mCitrine* was generated by recombining *PIL1p* (in pDONR221 P4-P1r) with *GUS* (in pDONR221 P1-P2) and *mCitrine* (in pDONR221 P2r-P3)[29] into the destination vector pH7m34GW. Plants were transformed with Agrobacterium tumefaciens GV3101 using the floral dip method[74], and segregation analysis of antibiotic resistance was used to isolate single-insertion homozygous lines.

### Growth conditions and hypocotyl measurements

*Arabidopsis thaliana* and *Nicotiana benthamiana* were sterilized, stratified in the dark at 4 °C, and grown vertically on 0.5X Linsmaier and Skoog media (LS; Caisson Laboratories) with 0.8% (w/v) micro-propagation type I agar (Caisson Laboratories) in LED chambers at 21 °C constant simulated white light (21WL, 70 μmol m$^{-2}$ s$^{-1}$, R/FR = 17.8) for 3 days and then either kept at 21WL or transferred to 30 °C (30WL, 70 μmol m$^{-2}$ s$^{-1}$, R/FR = 17.8) or 21 °C supplemented with far-red light (21FR, 70 μmol m$^{-2}$ s$^{-1}$, R/FR = 0.6) or 30 °C supplemented with far-red light (30FR, 70 μmol m$^{-2}$ s$^{-1}$, R/FR = 0.6), for an additional 3 days (Fig. 1a), unless described otherwise in the figure legend. For long day

(16 h day/8 h night) experiments, the light source was cool-white fluorescent bulbs. Base settings for Low Blue experiments were the same, with blue light reduced using filter #101 (Lee Filters, CA). For more details on light spectra, see Supplementary Data 1.

Measurements of the hypocotyl length were done on seedlings grown vertically and scanned at the indicated times. Measurements were done using ImageJ[75].

Tomato (*Solanum lycopersicum* cv M82, sp) was germinated on soil in a CONVIRON growth chamber under long-day conditions and LED white light (100 µmol m$^{-2}$ s$^{-1}$, Supplementary Data 1). Seedlings were grown in 21WL for 9 days and then moved to 21FR or 30WL or 30FR for an additional 12 days. The hypocotyls and epicotyls were measured with a ruler.

## Immunoblot analysis
Western blots were performed on protein extracts from 14 seedlings. The tissue was homogenized and boiled in 2X loading buffer with 1.8% β-mercaptoethanol (NuPAGE™ LDS sample buffer – ThermoFisher NP008) for 5 min. Extracts were run on Bis-tris 4–12% acrylamide gradient gel (Invitrogen, Carlsbad, CA, USA) and transferred to 0.45 µm nitrocellulose membrane by wet transfer. Primary antibodies used were anti-Myc, 9B11 (1:2000, catalog no. 2276, Cell Signaling Technology), anti-phyB[76] (1:2000), anti-Actin (1:50,000 catalog no. 2276, Sigma), and goat anti-mouse IgG (H + L)-HRP conjugate (1:5000, catalog no. 1706516, Bio-Rad). Actin was visualized by reprobing the membrane. The Sapphire Biomolecular Imager and AzureSpot Pro (Azure Biosystems) were used to quantify phyB protein levels.

## Imaging and epidermal cells length analysis
GUS (β-Glucuronidase) staining was performed as described previously[77]. Plant tissue was vacuum infiltrated in staining solution (25 mM phosphate buffer, pH 7, 0.25% Triton X-100, 2 mM potassium ferricyanide, 2 mM potassium ferrocyanide, 0.25 mM EDTA, 1 mg/ml 5 bromo-4 chloro 3 indolyl β-D-glucuronide X-Gluc), and incubated overnight at 37 °C. Tissue was then cleared in 95% ethanol and gradually brought to 50% ethanol with sequential washes and then to 50% glycerol.

For epidermal cell length measurements, seedlings were incubated in 50 µg/mL propidium iodide (PI) with 0.4% Triton X-100 for 10–20 min and then washed. Fluorescence images of the hypocotyl apex, middle and base segments were taken with a Zeiss LSM 710 confocal microscope, and the images were processed using ImageJ[75].

Scanning electron micrographs (SEM) images of the hypocotyl were taken by low vacuum scanning electron microscopy observations (LV-SEM) using a Zeiss Sigma VP-SEM. Live plants were transferred to an aluminum holder with a slab of agarose gel as substrate and immediately inserted into the microscope. After 3 min, the chamber pressure reached 5 Pa, and secondary electron images were generated using the VPSE detector at 15 kV using a 20 mm working distance. A representative image out of the five seedlings that were imaged per condition is shown.

## Hormone and chemical treatments
Seedlings were grown vertically on 0.5X LS plates on nylon mesh (100 mm, 44% open area, 40-inch-width ELKO filtering) in 21 °C constant simulated white light (21WL, 70 µmol m$^{-2}$ s$^{-1}$, R/FR = 17.8) for 3 days. After 3 days, the seedlings on the nylon mesh were transferred to plates containing picloram (P5575, Sigma), N-1-naphthylphthalamic acid (NPA, N6250, PhytoTechLABS), peo-IAA (HY-112730, MCE), geldanamycin (GDA, HY-15230, MCE), or DMSO control and grown under the indicated conditions for the indicated times.

## Real-time PCR
For quantitative (q) RT-PCR, RNA was extracted from 25 whole seedlings using the RNeasy Micro Kit (Qiagen). cDNA synthesis was

performed using Maxima first-strand cDNA synthesis kit (Thermo Fisher Scientific) with 1 µg of RNA. qRT-PCR analysis was carried out using CFX384 Real-Time PCR Detection System (Bio-Rad), with Premix Ex Taq II (TaKaRa, #RR820A). Levels of mRNA were calculated relative to *IPP2* as an internal control. Primers used for the qRT-PCR analysis are detailed in Supplementary Table 1.

## IAA and IAA-conjugate measurements
*Arabidopsis thaliana* seedlings were grown in 21WL (constant LED light) for 3 days and treated for 3 h in the indicated conditions to measure auxin and its conjugates. Seedlings were then snap-frozen in liquid nitrogen. The extraction, purification, and the LC-MS analysis of endogenous IAA and IAA metabolites were carried out according to Novak et al.[78]. Briefly, frozen samples (15–25 mg fresh weight) were homogenized using a MixerMill (Retsch GmbH, Haan, Germany) and extracted in ice-cold 1 ml 50 mM sodium phosphate buffer (pH 7.0) containing 1% sodium diethyldithiocarbamate, deuterium and 13C-labeled internal standards (5 pmol of [13C6]IAA, [13C6]IAA-glc, [13C6]IAAsp, and [13C6]IAGlu). The pH was adjusted to 2.7 with 1 M hydrochloric acid, and the samples were purified by solid phase extraction on hydrophilic-lipophilic balanced reversed-phase sorbent columns (Oasis® HLB, 1 cc/30 mg; Waters, Milford, MA, USA) conditioned with 1 ml methanol, 1 ml water, and 0.5 ml sodium phosphate buffer (pH 2.7). After sample application, the column was washed with 2 ml 5% methanol and then eluted with 2 ml 80% methanol. All eluates were evaporated at 37 °C to dryness in vacuo and dissolved in 40 µl of mobile phase prior to mass analysis. Ultra high-performance liquid chromatography (UHPLC-MS/MS) analysis was performed using a 1290 Infinity Binary LC System coupled to a 6490 Triple Quad LC/MS System (Agilent Technologies, Santa Clara, CA, USA), operating in multiple reaction monitoring (MRM) mode. The quantification was carried out in Agilent MassHunter Workstation quantitative analysis software (version B.05.02; Agilent Technologies) using stable isotope dilution method (Supplementary Data Set 5). Five independent biological replicates of each condition were used in these analyses.

## RNA-seq experiments and analysis
Two biological replicates of wild-type seedlings grown in constant LED white light (70 µmol m$^{-2}$ s$^{-1}$) at 21 °C (21WL) for 3 days and moved to 21FR, 30WL, 30FR, or left at 21WL (as in Fig. 1a). Samples (whole seedlings) were collected after 1 h, 3 h, 6 h, or 24 h in each of the conditions and snap-frozen in liquid nitrogen. Total RNA was isolated using the RNeasy Micro Kit (Qiagen). Stranded mRNA-seq libraries were prepared using the Illumina TruSeq Stranded mRNA Library Prep Kit according to the manufacturer's instruction. Libraries were sequenced with single-end 50 base-pair (bp) reads in the Illumina HiSeq 2500 System at the Salk Institute's Next Generation Sequencing (NGS) Core Facility. Raw reads were aligned to the TAIR10 genome using STAR (v2.5.4) (Dobin, Davis et al. 2013) with default parameters except–alignIntronMax 2000 and–outFilterMismatchNmax 2. Mapped reads were counted by HTSeq (v0.6.0) with default parameters except "-m intersection-strict -s reverse"[79] and differential expression was determined using edgeR (v3.16.5)[80,81]. Only genes with CPM (counts-per-million) values >1 in at least two samples were kept for differential expression analysis. A design matrix was set up where each treatment (light and temperature condition) at each time point is one group, and a generalized linear model was fitted to the read counts with this design matrix. For differential gene expression, contrasts were set up between 21WL and each of the light conditions (21FR, 30WL, or 30FR) at each time point.

Gene expression (cpm) and fold change can be found in Supplementary Data 2.

## ChIP-seq experiments and analysis
Two biological replicates of *PIF7:PIF7-4xMYC (pif457)* seedlings were grown in constant LED white light (70 µmol m$^{-2}$ s$^{-1}$) at 21 °C (21WL) for

4 days and were moved to 21FR, 30WL, 30FR, or were left at 21WL for 2 h. Afterward, seedlings were crosslinked and used for ChIP-seq analysis. ChIP-seq experiment was performed as previously described[82]. Monoclonal antibodies against MYC-tag Mouse (Cat#2276, Cell Signaling Technology) were used for immunoprecipitation. After elution, reverse crosslinking and DNA purification libraries were prepared using NEBNext® Ultra™ II DNA Library Prep Kit and amplified using 14 cycles in the thermocycler. Double size selection was made for control_30FR and replicates number 2 (aimed for 250 bp insert length), and the rest were size selected once (cleaning only small fragments). Multiplexed libraries were sequenced on an Illumina Nextseq2000. Bowtie2 (v2.2.5) was used with default parameters to map sequencing reads to the TAIR10 genome[83]. HOMER (v4.11.1) Findpeaks script[84] was used to call peaks using "-style factor -F 2 -P 0.001 -L 2 -LP 0.001" for control_21FR and replicates number one. "-F 4 -P 0.0001 -L 4 -LP 0.001" used for control_30FR and replicates number two. IP using anti-MYC antibody from wild-type (Col-0) grown in 21FR or 30FR were used as a background control (21FR used as a control for replicates one and 30FR used as a control for replicates two). The bedtools (v2.30.0) intersectBed function[85] was used to associate peaks with TAIR10 annotated genes within 2000 bp upstream of the transcription start site or with gene bodies. HOMER mergePeaks script was used to identify genes with binding in both replicates. The full list of genes and peaks can be found in Supplementary Data 4. Peaks were visualized using the IGV browser[86].

### Gene ontology analysis

Gene ontology enrichment analysis was performed using agriGO (v2.0) with fisher test and Yekutieli (FDR under dependency) for multi-test adjustment. TAIR genome locus (TAIR10 2017) was used as background[87].

### Reporting summary

Further information on research design is available in the Nature Research Reporting Summary linked to this article.

## Data availability

All data generated or analyzed during this study are included in this published article (and its supplementary information files). RNA-seq (GSE196725) and ChIP-seq (GSE205210) data have been deposited into the Gene Expression Omnibus: https://www.ncbi.nlm.nih.gov/geo/query/acc.cgi?acc=GSE196725 and https://www.ncbi.nlm.nih.gov/geo/query/acc.cgi?acc=GSE205210. Arabidopsis transgenic lines as well as plasmids generated during the current study are available from the corresponding author upon request. Source data are provided with this paper.

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

## Acknowledgements

We thank Ferenc Nagy (Plant Biology Institute, Hungary) for kindly providing the *35S:phyB-GFP*, *35S:phyB^Ser86Asp-YFP*, and *35S:phyB^Ser86Ala-YFP* seeds, Christian Fankhauser (University of Lausanne, Switzerland) for kindly providing the *cry1-304*, *cry1pif45*, *cry1pif457*, and *cry1phyB* seeds, Julin N. Maloof (University of California, Davis, USA) for kindly providing the *yuc2589* seeds, William M. Gray (University of Minnesota, USA) for kindly providing the *eta3* seeds, Akira Nagatani (Kyoto University, Japan) for kindly providing phyB antibody, Meng Chen (University of California, Riverside, USA) for the help with the phyB western blot and Leo Andrade (Salk Biophotonics Core) for helping with the SEM. J.C. is an investigator of the Howard Hughes Medical Institute. The study was supported by a grant from NIH (5R35GM122604-05_05) to J.C. and by the Howard Hughes Medical Institute to J.C. and by grants from the Knut and Alice Wallenberg Foundation (KAW 2016.0341 and KAW 2016.0352) and the Swedish Governmental Agency for Innovation Systems (VINNOVA 2016-00504) to K.L. and O.N. Y.B. was funded by EMBO Fellowship (ALTF 785-2013) and BARD (FI-488-13). B.C.W. was supported by EMBO Fellowship (ALTF 1514-2012), the Human Frontier Science Program (LT000222/2013-L), and Salk Pioneer Postdoctoral Endowment Fund. A.S. was supported by the Salk Pioneer Postdoctoral Endowment Fund. The NGS Core Facility of the Salk Institute is supported by funding from NIH-NCI (CCSG: P30 014195), the Chapman Foundation, and the Helmsley Charitable Trust. The Salk Institute Waitt Advanced Biophotonics Core is supported by funding from NIH-NCI CCSG: P30 014195, the Waitt Foundation, and the Chan-Zuckerberg Initiative Imaging Scientist Award.

## Author contributions

Y.B., B.C.W., A.S., and J.C. designed the research and wrote the article. J.C. supervised. Y.B. performed all the experiments and analyzed the data. B.C.W. generated *pif* and *phyB pif* mutant combinations and performed the ChIP-seq experiment. K.L. and O.N. measured the IAA and IAA metabolites.

## Competing interests

The authors declare no competing interests.
