## [Peer Review File · Nature Communications]

PIF7 is a master regulator of thermomorphogenesis in shadeREVIEWER COMMENTS

Reviewer #1 (Remarks to the Author):

The manuscript by Burko et al. reports that PIF7 plays the major and dominant role under shade combined with warm temperature. Authors demonstrate that a combination of shade (low R/FR) and warm temperature synergistically enhance hypocotyl elongation (Fig. 1). As PIF4 and PIF7 have been known as key regulators in thermomorphogenesis and shade, authors tested which PIFs are required for this synergistic response of warm temperature and shade using a combination of pif mutants. This genetic analyses revealed that PIF7 plays a dominant role under combined warm temperature and shade condition, while PIF4 has a minor role (Fig. 2). Authors further showed that under low blue light condition, PIF4 plays relatively more major role than PIF7 (Fig. 3A) and PIF7 functions in downstream of blue light photoreceptor CRY1 as well as phyB (Fig. 3B, C). PIF7 activity was not significantly changed in this combined condition as demonstrated with dephosphorylated western analysis and expression levels of PIF7 targets using RNA-seq analysis (Fig. 4). Authors found that YUC8 auxin biosynthetic gene expression is up-regulated significantly with shade, but marginally with warm temperature (Fig. 5). Free auxin (IAA) level was not changed, but the several IAA-conjugates showed different levels (Sup Fig 6).

Regulation of auxin sensitivity through HSP90 was not involved in this synergistic response (Sup Fig 8). However, auxin analog paclobarot treatment enhances hypocotyl growth stronger at 30WL condition (Fig 6). Together, authors concluded that a combination of warm temperature and shade synergistically enhance hypocotyl growth via PIF7 as a major role in regulating auxin biosynthesis/sensitivity and via unknown factors activated by warm temperature.

Overall, the major results about the dominant role of PIF7 in synergistic response of hypocotyl growth by warm temperature and shade are significant discoveries in the field of light and temperature signaling. The data presented in this manuscript generally supports well their conclusion and major discovery with sufficient genetic evidence. However, the proposed physiological and molecular mechanisms by which PIF7 specifically play a role in this combined condition may not fully explain the hypocotyl growth phenotype. I have several comments regarding this.

1. Authors conclude that PIF7 activity under shade is not increased by ambient temperature, based on expression of direct PIF7 targets. It seems like that the ChIP experiment for PIF7 was done in different condition (only 21C) in previous publication (Willige et al., 2021 Nat Gen 53:955). It is possible that PIF7 may have different targets specifically in 30FR condition. To avoid over-simplified conclusion without this dataset, I think that authors could, at least, provide more useful information. One thought is 1) identifying specific group of differentially regulated genes in 30FR condition, 2) check PIF7-dependent expression (qPCR) or promoter binding (ChIP-qPCR if G-boxes present) for some of representative genes (functionally relevant) in 30FR condition. It is also possible that PIF7 may regulate some of key genes necessary for synergistic response as heterodimer with other TFs such as bHLH48/60 (Yang et al., 2021 Cell Reports, 35:109054). Thus, more analysis using RNA-seq data may provide more valuable information about the action mechanism of PIF7.

2. The data regarding auxin biosynthesis and sensitivity is valuable information telling physiological mechanisms by which PIF7 plays a major role in auxin biosynthesis and/or sensitivity under combined warm temperature and shade. Authors reported that they did not observe significant change in IAA in any of their conditions. However, increased IAA level under shade or warm temperature has been reported previously, but authors reported similar (or at least non statistically significant) levels under 21FR and 30FR. I feel that this analysis needs to be reexamined. It seems like that at least shade treatment marginally increased free IAA level (Sup. 6A). Is this statistically significant? Also data in Sup Fig 6A and B needs one-way ANOVA analysis for the comparison of all variables.

3. Authors tested the possibility of different auxin sensitivity with HSP90. I feel their conclusion with negative data is valuable. But there are other factors affecting complex auxin signaling and sensitivity

via PIF4 (Pucciariello et al., 2018, PNAS 115:5612-5617). This involves TIR1/AFBs, ARF transcription factors, miR393, etc. Does a combined warm temperature and shade condition regulate expression of any of these factors?

Alternatively, DR5:GUS line or DII fluorescence can be used to determine whether auxin sensitivity is indeed involved in PIF7-dependent synergistic regulation.

4. In line 170-171, authors suggest that part of the response to 30WL in phyB is PIF-independent. It is accurate to say PIF4/5-independent. Also, isn't this phyB-independent as well? Existence of phyB-PIF independent thermomorphogenesis has been recently reported by Vu et al. 2021 (Nat Commun 12:2842).

5. Is phyB protein level altered in a combined warm temperature and shade condition? Considering the inhibitory role of PHYB on PIF7 activity (Willige et al. 2021 Nat Genet), and increasing phyB protein level in warm temperature (Hahm et al. 2020 Nat Commun), it would be valuable to know phyB protein level in combined condition.

6. Any explanation about no clear shade phenotype of pif4 and pif4pif5 mutant in 21FR condition?

7. Although I like current model in Fig 6C, it might be informative if it can show how CRY1 and PHYB differentially regulate PIF4 and PIF7 under different condition to capture authors' explanation on CRY1-PIF4 regulation in R/FR, and PIF4/PIF7 ratio, etc. Also, if the role PIF4 in shade with normal temperature is clear, PIF4 should be included in a model with shade condition.

Reviewer #2 (Remarks to the Author):

Growth of plants is strongly affected by light conditions such as sunlight vs. canopy shade and temperature. The light spectrum in canopy shade is characterised by a low red:far-red light ratio (R:FR) that inactivates phyB and thereby results in increased growth in shade-intolerant species. At the same time, growth is also enhanced at elevated temperature, leading to better air circulation and cooling of leaves. Members of the PIF family are key factors for these responses with PIF7 playing a dominant role for enhancing growth in response to shade conditions and PIF4/PIF5 promoting growth at elevated temperature. Under natural conditions, plants are not unlikely to experience canopy shade and elevated temperature at the same time but previous studies only investigated the effect of either shade or elevated temperature but not the combination of both. The study by Burko and co-authors addresses this important question. They very nicely and convincingly show that the effect of shade and elevated temperature is additive, i.e. growth of seedlings exposed to shade and elevated temperature is dramatically increased compared to seedlings exposed to only shade or elevated temperature. This effect is almost fully abolished in mutants lacking functional PIF7, while lack of PIF4 and PIF5 does not affect the response to combined shade and elevated temperature. Given the essential role of PIF4/5 in thermomorphogenesis in sunlight, this finding is very surprising and interesting. Overall, this is a very well-written manuscript presenting high quality data. Even though the authors could not identify the exact molecular mechanism underlying the synergistic effect of shade and elevated temperature, the manuscript presents new and interesting data that are well worth being published in Nature Communications. I do not have any major concerns; minor comments are listed below.

Line 174ff:

The authors compare pif457 expressing PIF7:PIF7-MYC with pif457 to test if PIF7 plays a role in 21FR or 30FR... and then say that this line phenocopies pif45. I agree with the conclusion but isn't it a bit unusual to put it this way, i.e. wouldn't it be more intuitive to compare pif457 and pif45 to get information on the function of PIF7 and then say that the lack of PIF7 in pif457 can be complemented

by the PIF7:PIF7-MYC transgene (which is an important control since the authors are using this line for testing PIF7 phosphorylation).

Line 485:

What does "21SH" mean?

Line 519:

The authors write "... Another possibility is that cell wall rigidity, fluidity, or permeability may be regulated directly by temperature, potentially leading to exaggerated elongation at 30FR". Do they really mean fluidity of the cell wall? Or do they mean fluidity of the cell membrane?

Fig. 3C, Fig. S3C:

The phyB mutant in 30WL has much longer hypocotyls than at 30FR which I find surprising. Do the authors have any potential explanation?

Suppl. Data 1 (light conditions):

I appreciate that the authors provide light spectra. However, the labelling is unclear to me, i.e. light spectra in column B in the first sheet do not correspond to labels of tabs showing the different spectra. Please also double-check the unit of the y-axis; I think the plots show spectral photon flux, not photon flux/fluence rate, and therefore the units should be $\mu\text{mol m}^{-2} \text{s}^{-1} \text{nm}^{-1}$ ($\mu\text{E nm}^{-1}$).

Reviewer #3 (Remarks to the Author):

Burko et al described the synergistic impact on plant development by shade and high ambient temperature. Previously, the impact of shade and high ambient temperature has been studied separately by many groups. However, in nature plants are often exposed to these stimuli simultaneously as opposed to a single stimulus. Thus, understanding how plants respond to these combined stimuli is important. These authors found that plants display synergistic phenotype when exposed to both stimuli. They also found that PIF7 is a master regulator under these conditions. However, after exhaustive testing of many factors, hormones, they still didn't have a molecular mechanism of the synergistic phenotype. Here are a few minor suggestions:

1. I am surprised that they didn't find any synergistic gene expression in RNAseq study. From the growth assays as shown in Fig. 1A, F, the optimum temperature for growth appears to be 28, although there is no statistically significant difference between 28 and 30. But there is a downward trend at 30. I wonder performing all their assays at 30 might have masked some missing factors.

2. line 221-223, the authors said

"We found that the response to warm temperature under WL (30WL) in phyB mutants was similar to wild-type seedlings. However, elongation growth in response to low R/FR at either 21°C or 30°C was compromised in phyB mutants (Fig. 3C, Supplementary Fig. 3C,D)."

But the fig 3C shows phyB mutants have longer hypocotyls than WT and show similar hypocotyls compared to WT in low R/FR at either 21°C or 30°C. It doesn't match their description.

3. In line 309, 485, 518, there might be a typo, 30SH or 21SH?

REVIEWER COMMENTS

Reviewer #1 (Remarks to the Author):

The manuscript by Burko et al. reports that PIF7 plays the major and dominant role under shade combined with warm temperature. Authors demonstrate that a combination of shade (low R/FR) and warm temperature synergistically enhance hypocotyl elongation (Fig. 1). As PIF4 and PIF7 have been known as key regulators in thermomorphogenesis and shade, authors tested which PIFs are required for this synergistic response of warm temperature and shade using a combination of pif mutants. This genetic analyses revealed that PIF7 plays a dominant role under combined warm temperature and shade condition, while PIF4 has a minor role (Fig. 2). Authors further showed that under low blue light condition, PIF4 plays relatively more major role than PIF7 (Fig. 3A) and PIF7 functions in downstream of blue light photoreceptor CRY1 as well as phyB (Fig. 3B, C). PIF7 activity was not significantly changed in this combined condition as demonstrated with dephosphorylated western analysis and expression levels of PIF7 targets using RNA-seq analysis (Fig. 4). Authors found that YUC8 auxin biosynthetic gene expression is up-regulated significantly with shade, but marginally with warm temperature (Fig. 5). Free auxin (IAA) level was not changed, but the several IAA-conjugates showed different levels (Sup Fig 6).

Regulation of auxin sensitivity through HSP90 was not involved in this synergistic response (Sup Fig 8). However, auxin analog paclobutrazol treatment enhances hypocotyl growth stronger at 30WL condition (Fig 6). Together, authors concluded that a combination of warm temperature and shade synergistically enhance hypocotyl growth via PIF7 as a major role in regulating auxin biosynthesis/sensitivity and via unknown factors activated by warm temperature.

Overall, the major results about the dominant role of PIF7 in synergistic response of hypocotyl growth by warm temperature and shade are significant discoveries in the field of light and temperature signaling. The data presented in this manuscript generally supports well their conclusion and major discovery with sufficient genetic evidence. However, the proposed physiological and molecular mechanisms by which PIF7 specifically play a role in this combined condition may not fully explain the hypocotyl growth phenotype. I have several comments regarding this.

We thank the reviewer for their kind words, the positive feedback, and helpful suggestions on how to improve the manuscript.

1. Authors conclude that PIF7 activity under shade is not increased by ambient temperature, based on expression of direct PIF7 targets. It seems like that the ChIP experiment for PIF7 was done in different condition (only 21C) in previous publication (Willige et al., 2021 Nat Gen 53:955). It is possible that PIF7 may have different targets specifically in 30FR condition. To avoid over-simplified conclusion without this dataset, I think that authors could, at least, provide more useful information. One thought is 1) identifying specific group of differentially regulated genes in 30FR condition, 2) check PIF7-dependent expression (qPCR) or promoter binding (ChIP-qPCR if G-boxes present) for some of representative genes (functionally relevant) in 30FR condition.

We thank the reviewer for this comment and share the reviewer's concern; therefore, we decided to perform PIF7 ChIP-seq in all tested conditions. The ChIP-seq revealed no differences in PIF7 binding in 21FR and 30FR conditions. We found that PIF7 can bind to the DNA at 30WL (and not at 21WL). Interestingly, PIF7-MYC was able to bind to its DNA targets in 30WL, but was unable to promote growth in the *pif457* triple mutant background in the same growth condition. Therefore, we concluded that warm temperature is sufficient to promote PIF7 binding to the DNA but insufficient to promote PIF7-dependent gene regulation. To our best knowledge, this is the first time this has been shown.

We added these results in the text (lines 205-217), in Fig 4g,h and as new Sup. Data set #4. We also changed the list of the genes used for Fig. 4f to the top 50 genes that we identified as bound by PIF7 in 30FR.

It is also possible that PIF7 may regulate some of key genes necessary for synergistic response as heterodimer with other TFs such as bHLH48/60 (Yang et al., 2021 Cell Reports, 35:109054). Thus, more analysis using RNA-seq data may provide more valuable information about the action mechanism of PIF7.

We agree with the reviewer that PIF7 might regulate key genes as a heterodimer with other TFs. To address the possibility, we compare the expression of PIF7/bHLH60 co-bound genes and PIF7-regulated genes (described in Yang et al., 2021 Cell Reports). We hypothesize that if PIF7 specificity in 30FR will be determined by its activity as a heterodimer with bHLH60 we should see differences in the expression of their common targets. However, as shown in the boxplot below, there is no obvious change in their expression in any of our growth conditions. It is still possible that specific genes are regulated by PIF7 bHLH48/60 heterodimers and contribute to the synergistic response, but we could not detect them. We can include this figure in the supplement if requested.

2. The data regarding auxin biosynthesis and sensitivity is valuable information telling physiological mechanisms by which PIF7 plays a major role in auxin biosynthesis and/or sensitivity under combined warm temperature and shade. Authors reported that they did not

observe significant change in IAA in any of their conditions. However, increased IAA level under shade or warm temperature has been reported previously, but authors reported similar (or at least non statistically significant) levels under 21FR and 30FR. I feel that this analysis needs to be reexamined. It seems like that at least shade treatment marginally increased free IAA level (Sup. 6A). Is this statistically significant? Also data in Sup Fig 6A and B needs one-way ANOVA analysis for the comparison of all variables.

We thank the reviewer for this helpful suggestion, and we agree that there is a slight increase in the IAA level in response to low R/FR, as can also be seen with the IAA conjugates. However, this difference is not significant when we apply one-way ANOVA and Tukey HSD test.

We added the one-way ANOVA result to Sup. Fig. 6a, and the description of the full statistics to the source data file (including the t-tests).

In addition, the statistics present in Sup. Fig. 6b was based on the edgeR analysis. We made sure that this was clearly noted in the figure legends. The full report of this analysis can be found in the source data file.

3. Authors tested the possibility of different auxin sensitivity with HSP90. I feel their conclusion with negative data is valuable. But there are other factors affecting complex auxin signaling and sensitivity via PIF4 (Pucciariello et al., 2018, PNAS 115:5612-5617). This involves TIR1/AFBs, ARF transcription factors, miR393, etc. Does a combined warm temperature and shade condition regulate expression of any of these factors?

Thank you for pointing this out. We could not detect the expression of miR393 in our RNA-seq data. The expression of miR393 targets (also regulated by PIF4), TIR1, AFB2, 3, and 5, did not change in any of our growing conditions. The same was true for other PIF4 targets such as ARF6, TAA1, and CYP79B2.

Alternatively, DR5:GUS line or DII fluorescence can be used to determine whether auxin sensitivity is indeed involved in PIF7-dependent synergistic regulation.

We added *DR5:GUS* staining to Sup Fig 6e and described it in the text line 290. The pattern of the *DR5:GUS* was similar to *SAUR19:GUS* (higher in 21FR and 30FR compared to 30WL and 21WL). Suggests that auxin sensitivity is involved in the PIF7-dependent synergistic growth at 30FR in the same manner as in 21FR.

4. In line 170-171, authors suggest that part of the response to 30WL in phyB is PIF-independent. It is accurate to say PIF457-independent. Also, isn't this phyB-independent as well? Existence of phyB-PIF independent thermomorphogenesis has been recently reported by Vu et al. 2021 (Nat Commun 12:2842).

Thank you for the suggestion and for drawing our attention to the Vu et al. 2021 Nature Commun paper. We changed the text accordingly, cited the paper, and added that phyB-PIF independent thermomorphogenesis had been shown before. See Lines 138-139

5. Is phyB protein level altered in a combined warm temperature and shade condition? Considering the inhibitory role of PHYB on PIF7 activity (Willige et al. 2021 Nat Genet), and increasing phyB protein level in warm temperature (Hahm et al. 2020 Nat Commun), it would be valuable to know phyB protein level in combined condition.

This is an interesting point. We obtained the native antibodies and performed the western blot for phyB. We were unable to detect a change in phyB abundance in any of our conditions. We added these results in Sup Fig 3e,f and in the text lines 180-181. Additionally, we would like to mention that there are contradictory reports about phyB accumulation in warm temperatures (higher in Hahm et al. 2020 Nat Commun and lower in Lee et al. 2021 PLOS Genetics).

6. Any explanation about no clear shade phenotype of pif4 and pif4pif5 mutant in 21FR condition?

This is a very interesting point that we still don't fully understand. It might be due to different interactions between PIF7 and PIF4,5 with the photoreceptors. In addition, It was shown before that *pif45* has a subtle effect on hypocotyl elongation in 21FR (Li et al. 2012 Genes & Development). While no significant differences were found in our study between WT and *pif45* or *pif4* when one-way ANOVA and Tukey HSD test was used, a t-test did find a significant difference between WT and *pif45* at 21FR (presented in the source data).

7. Although I like current model in Fig 6C, it might be informative if it can show how CRY1 and PHYB differentially regulate PIF4 and PIF7 under different condition to capture authors'

explanation on CRY1-PIF4 regulation in R/FR, and PIF4/PIF7 ratio, etc. Also, if the role PIF4 in shade with normal temperature is clear, PIF4 should be included in a model with shade condition.

Thank you for this suggestion. We added cry1 and phyB to our model. We decided to keep it general as we still do not fully understand the different regulations of cry1 and phyB over PIF7 and PIF4, especially given our results that PIF7 works downstream to cry1.

Reviewer #2 (Remarks to the Author):

Growth of plants is strongly affected by light conditions such as sunlight vs. canopy shade and temperature. The light spectrum in canopy shade is characterized by a low red:far-red light ratio (R:FR) that inactivates phyB and thereby results in increased growth in shade-intolerant species. At the same time, growth is also enhanced at elevated temperature, leading to better air circulation and cooling of leaves. Members of the PIF family are key factors for these responses with PIF7 playing a dominant role for enhancing growth in response to shade conditions and PIF4/PIF5 promoting growth at elevated temperature. Under natural conditions, plants are not unlikely to experience canopy shade and elevated temperature at the same time but previous studies only investigated the effect of either shade or elevated temperature but not the combination of both. The study by Burko and co-authors addresses this important question. They very nicely and convincingly show that the effect of shade and elevated temperature is additive, i.e. growth of seedlings exposed to shade and elevated temperature is dramatically increased compared to seedlings exposed to only shade or elevated temperature. This effect is almost fully abolished in mutants lacking functional PIF7, while lack of PIF4 and PIF5 does not affect the response to combined shade and elevated temperature. Given the essential role of PIF4/5 in thermomorphogenesis in sunlight, this finding is very surprising and interesting. Overall, this is a very well-written manuscript presenting high quality data. Even though the authors could not identify the exact molecular mechanism underlying the synergistic effect of shade and elevated temperature, the manuscript presents new and interesting data that are well worth being published in Nature Communications. I do not have any major concerns; minor comments are listed below.

We thank the reviewer for the kind words and encouraging feedback about our manuscript.

Line 174:

The authors compare pif457 expressing PIF7:PIF7-MYC with pif457 to test if PIF7 plays a role in 21FR or 30FR... and then say that this line phenocopies pif45. I agree with the conclusion but isn't it a bit unusual to put it this way, i.e. wouldn't it be more intuitive to compare pif457 and pif45 to get information on the function of PIF7 and then say that the lack of PIF7 in pif457 can be complemented by the PIF7:PIF7-MYC transgene (which is an important control since the authors are using this line for testing PIF7 phosphorylation).

Thank you for pointing this out. We made the necessary change in the text lines 145-148

Line 485:

What does "21SH" mean?

We appreciate the reviewer's careful reading of the text. 21SH is a typo of 21FR. We made the necessary correction.

Line 519:

The authors write "... Another possibility is that cell wall rigidity, fluidity, or permeability may be regulated directly by temperature, potentially leading to exaggerated elongation at 30FR". Do they really mean fluidity of the cell wall? Or do they mean fluidity of the cell membrane?

We changed the text to cell membrane fluidity.

Fig. 3C, Fig. S3C:

The phyB mutant in 30WL has much longer hypocotyls than at 30FR which I find surprising. Do the authors have any potential explanation?

This is an interesting point. The phyB mutant was previously described as having a shorter hypocotyl than wild-type in low R/FR conditions, which we observed in both 21FR and 30FR. This phenotype depends on the PAR, the age of the seedlings, and the ratio between R/FR, as described in Martínez-García et al. 2014 PloS one. While we still don't fully understand these phenomena, some explanations suggest that it is due to the phyA repression activity in low R/FR conditions. At the same time, the repression activity of PIL1 and HFR1 could also be part of this mechanism.

Suppl. Data 1 (light conditions):

I appreciate that the authors provide light spectra. However, the labelling is unclear to me, i.e. light spectra in column B in the first sheet do not correspond to labels of tabs showing the different spectra. Please also double-check the unit of the y-axis; I think the plots show spectral photon flux, not photon flux/fluence rate, and therefore the units should be $\mu\text{mol m}^{-2} \text{s}^{-1} \text{nm}^{-1}$ ($\mu\text{E nm}^{-1}$).

We changed the labeling so they would fit the excel sheet names and added the temperatures used in each of the chambers. In addition, we change μE to $\mu\text{mol m}^{-2} \text{s}^{-1}$ throughout the manuscript, Figures and Supplementary Figures and Data.

Reviewer #3 (Remarks to the Author):

Burko et al described the synergistic impact on plant development by shade and high ambient temperature. Previously, the impact of shade and high ambient temperature has been studied

separately by many groups. However, in nature plants are often exposed to these stimuli simultaneously as opposed to a single stimulus. Thus, understanding how plants respond to these combined stimuli is important. These authors found that plants display synergistic phenotype when exposed to both stimuli. They also found that PIF7 is a master regulator under these conditions. However, after exhaustive testing of many factors, hormones, they still didn't have a molecular mechanism of the synergistic phenotype. Here are a few minor suggestions:

We thank the reviewer for the encouraging feedback about our manuscript.

1. I am surprised that they didn't find any synergistic gene expression in RNAseq study. From the growth assays as shown in Fig. 1A, F, the optimum temperature for growth appears to be 28, although there is no statistically significant difference between 28 and 30. But there is a downward trend at 30. I wonder performing all their assays at 30 might have masked some missing factors.

Thank you for pointing this out. We agree that it is surprising that we didn't find many differences in the RNA-seq data. While we could not exclude the option that this is due to the temperature we used, we strongly believe that if the synergistic response could be explained by gene expression, we were able to detect it also in 30°C:

First, the important role of PIF7 that we see in 30FR can also be observed in 28FR (Fig. 2a and Sup. Fig. 3b). Second, we see the typical and previously reported shade avoidance and auxin responses in 21FR as well as in 30FR, in our gene expression datasets. Therefore, we do not think that the additional 2 degree mask any potential synergism.

In addition, we performed further analysis of the RNA-seq data, specifically looking for genes that change their expression in response in 30FR but not in 21FR or 30WL (Sup Fig. 7b-f). While we did find a few genes that were changed specifically in 30FR in comparison to all other conditions, we don't think they can explain the synergistic elongation in 30FR for the following reasons:

1. They are only found in one-time point (not even one gene changes in all-time points), meaning there is no new set of genes specifically regulated in 30FR.
2. We performed ChIP-seq in all of our conditions and found that PIF7 bound to the same genes in 21FR and 30FR. In addition, the majority of the 30FR specific genes were not directly regulated by PIF7.
3. Most of the genes that did differential expressed and bound by PIF7, bound in both 21FR and 30FR (Sup Fig. 7d).

Nevertheless, we added the information in the discussion that these genes might explain the synergistic response (lines 306, 421-423).

2. line 221-223, the authors said

"We found that the response to warm temperature under WL (30WL) in phyB mutants was similar to wild-type seedlings. However, elongation growth in response to low R/FR at either 21°C or 30°C was compromised in phyB mutants (Fig. 3C, Supplementary Fig. 3C,D)."
But the fig 3C shows phyB mutants have longer hypocotyls than WT and show similar hypocotyls compared to WT in low R/FR at either 21°C or 30°C. It doesn't match their description.

We thank the reviewer for this comment. Since phyB mutant is already longer than the wild type in 21WL, it is hard to assess its response to 30WL based on the hypocotyl length. We actually wanted to describe that the relative growth response (length at 30WL / length at 21WL) was similar between wild type and the phyB mutant (as shown in Sup Fig. 3d). To clarify this point, we changed this sentence to "We found that the relative response ... was similar to wild-type seedlings" (lines 178-179).

3. In line 309, 485, 518, there might be a typo, 30SH or 21SH?

We appreciate the reviewer's careful reading of the text. We made the necessary correction.

REVIEWERS' COMMENTS

Reviewer #1 (Remarks to the Author):

The revised manuscript by Burko et al. has been well improved. The authors performed additional key experiments (ChIP-seq in all conditions) and discovered additional finding that warm temperature induced DNA-binding activity, not transactivation activity of PIF7. The authors satisfactorily addressed all of my comments and made all the necessary changes to the manuscript. I have no additional concerns.

Reviewer #2 (Remarks to the Author):

I thank the authors for their response. They have fully addressed all my comments.

Reviewer #3 (Remarks to the Author):

My concerns have largely been addressed. Although no clear mechanistic details for the synergistic phenotype has been shown, the manuscript has improved with additional data and interpretation.